# Xcr1+ type 1 conventional dendritic cells are essential mediators for atherosclerosis progression

Tianhan Li[1†], Liaoxun Lu[2†], Juanjuan Qiu[2], Xin Dong[2], Le Yang[2], Kexin He[2], Yanrong Gu[2], Binhui Zhou[2], Tingting Jia[2], Toby Lawrence[3], Marie Malissen[3], Guixue Wang[4], Rong Huang[2], Hui Wang[2], Bernard Malissen[3], Yinming Liang[2,5]*, Lichen Zhang[2]*

[1]School of Basic Medicine, Xinxiang Medical University, Xinxiang, China; [2]School of Medical Technology, Xinxiang Medical University, Xinxiang, China; [3]Centre d'Immunologie de Marseille-Luminy, INSERM, CNRS, Aix-Marseille Université, Marseille, France; [4]Key Laboratory for Biorheological Science and Technology of Ministry of Education, State and Local Joint Engineering Laboratory for Vascular Implants, Bioengineering College of Chongqing University, Chongqing, China; [5]Hunan Key Laboratory of Disease Models for Integrative Medicine,Center of Disease Model and Immunology, Hunan Academy of Chinese Medicine, Changsha, China

*For correspondence:
yinming.liang@gris.org.cn (YL);
zhanglichen@gris.org.cn (LZ)

†These authors contributed equally to this work

Competing interest: The authors declare that no competing interests exist.

## eLife Assessment

This manuscript by Li, Lu et al., presents **important** findings on the role of cDC1 in atherosclerosis and their influence on the adaptive immune system. Using Xcr1Cre-Gfp Rosa26LSL-DTA ApoE-/- mouse models, these data **convincingly** reveal an unexpected, non-redundant role of the XCL1-XCR1 axis in mediating cDC1 contributions to atherosclerosis.

**Abstract** Atherosclerosis is characterized by lipid accumulation within plaques, leading to foam cell formation and an inflammatory response within the aortic lesions. Lipid disorders have been extensively investigated, however, the cellular and molecular mechanisms that trigger the inflammatory response in atherosclerotic plaques remain far from being fully understood. Xcr1+ cDC1 cells are newly identified antigen-presenting cells in activating immune cells. However, the role of cDC1 cells in atherosclerosis development remains highly controversial. We first confirmed the presence of cDC1 within human atherosclerotic plaques and discovered a significant association between the increasing cDC1 numbers and atherosclerosis progression in mice. Subsequently, we established $Xcr1^{Cre-Gfp} Rosa26^{LSL-DTA} Apoe^{-/-}$ mice, a novel and complex genetic model, in which cDC1 was constitutively depleted in vivo during atherosclerosis development. Intriguingly, we observed a notable reduction in atherosclerotic lesions in hyperlipidemic mice, alongside suppressed T cell activation of both CD4+ and CD8+ subsets in the aortic plaques. Notably, aortic macrophages and serum lipid levels were not significantly changed in the cDC1-depleted mice. Single-cell RNA sequencing revealed heterogeneity of Xcr1+ cDC1 cells across the aorta and lymphoid organs under hyperlipidemic conditions. As Xcr1 is the sole receptor for Xcl1, we next explored to target Xcr1+ cDC1 cells via Xcl1 by establishing $Xcl1^{-/-}Apoe^{-/-}$ mice. $Xcl1^{-/-}Apoe^{-/-}$ mice exhibited decreased atherosclerotic plaque formation and reduced aortic cDC1 accumulation, indicating that Xcl1 contributes to cDC1-mediated atherosclerotic lesion development. Our results reveal crucial roles of cDC1 in atherosclerosis progression and provide insights into the development of immunotherapies by targeting cDC1 through Xcl1.

## Introduction

Atherosclerosis is a chronic inflammatory disease that affects the large and medium-sized arteries, and it is the leading cause of cardiovascular mortality and morbidity worldwide (*Björkegren and Lusis, 2022*). Atherosclerosis is a complex pathological process during which lipid-laden phagocytes such as macrophages, along with lymphocytes and dendritic cells, accumulate in the plaque (*Hansson and Hermansson, 2011*). Mounting evidence from animal models and human studies demonstrates involvement of both the innate and adaptive immunity in the development of atherosclerosis (*Lusta et al., 2023*; *Hansson et al., 2006*; *Hasham and Pillarisetti, 2006*). In healthy arteries, DCs are typically present in the subendothelial space and work in the front line of immune surveillance (*Steinman et al., 2003*; *Ma-Krupa et al., 2004*), and a small portion of resident vascular DCs are localized in the adventitia. Various studies have demonstrated that DCs accumulate in the advanced atherosclerotic lesions in both humans and animal models (*Liu et al., 2008*; *Jongstra-Bilen et al., 2006*; *Bobryshev and Lord, 1995*; *Kawahara et al., 2007*). Recently, several phenotypically and functionally distinct vascular DCs have been identified in mouse and human atherosclerotic lesions (*Zernecke et al., 2023*). Xcr1$^+$ type 1 conventional dendritic cells (Xcr1$^+$ cDC1) (also known as CD8$^+$ DC or CD103$^+$ DCs) has been recognized for their role in cross-presentation and activating naive CD8$^+$ T cells, and are critical for anti-viral and anti-tumor immune responses (*Eisenbarth, 2019*; *Bayerl et al., 2023*). Owing to the challenges associated with obtaining a model for specific cDC1 depletion in vivo, previous studies investigating the role of cDC1 cells in atherosclerosis using various knockout mice have produced conflicting results (*Legein et al., 2015*; *Gil-Pulido et al., 2017*; *Choi et al., 2011*; *Clément et al., 2018*; *Haddad et al., 2017*). It is important to note that previous Cre lines developed to achieve specific Cre expression in cDC1s still led to leakiness in recombinase activity outside of cDC1s, as demonstrated in previous publications (*Wohn et al., 2020*; *Mundt et al., 2019*). Moreover, scant mechanistic understanding regarding the regulation of cDC1 activity during atherosclerosis has been reported. Therefore, developing a more specific mouse model for cDC1 is essential for elucidating its precise role in atherosclerosis. Here, we established a novel Cre line *Xcr1$^{Cre-Gfp}$* mice and did not observe any leakiness of Cre activity, and further experiments revealed that specific depletion of cDC1 attenuates atherosclerosis development by inhibiting both CD4$^+$ and CD8$^+$ T cell activation in hyperlipidemic *Xcr1$^{Cre-Gfp}$ Rosa26$^{LSL-DTA}$ Apoe$^{-/-}$* mice.

The origin of phagocytes within the atherosclerotic plaques has been controversial. Studies have shown that the major contribution of macrophages in the aortic plaque is recruited from the circulation, while other studies have emphasized the significance of local proliferation (*Robbins et al., 2013*; *Potteaux et al., 2011*). Tissue-resident dendritic cells inside the plaques have been well documented (*Yilmaz et al., 2006*; *Alberts-Grill et al., 2013*), regardless of the fact that the contribution and origin of Xcr1$^+$ cDC1 have not been appropriately addressed. To understand the origin of Xcr1$^+$ cDC1 inside the plaque, we performed bone marrow transplantation using *Xcr1$^{Cre-Gfp}$ Rosa26$^{LSL-DTA}$* mice as donors and found that in *Apoe$^{-/-}$* recipient mice placed on a high-fat diet (HFD), cDC1 cells were completely absent in the aortas. Therefore, our experiments have elucidated that cDC1 cells in the atherosclerotic plaques originate from the bone marrow. Further single-cell RNA sequencing illustrated that notable differences in Xcr1$^+$ cDC1 cells are discovered among the aorta, spleen, and lymph nodes. *Ccr2*, *Sept3*, and *Cldnd1* were highly expressed in Xcr1$^+$ cDC1 cells from aorta compared with that from spleen and lymph nodes.

We further aimed to ascertain whether cDC1 can be modulated by targeting the chemokine Xcl1, which is a ligand of Xcr1. Considering the vital role that Xcr1$^+$ cDC1 plays in the progression of atherosclerosis and the therapeutic potential of manipulating their functions via chemokines (*Yan et al., 2021*), comprehending the feasibility of targeting Xcl1 holds considerable promise for developing novel therapeutic approaches to halt the advancement of atherosclerosis. We conducted further studies using *Xcl1$^{-/-}$ Apoe$^{-/-}$* double knockout mice to assess the pathological grade of atherosclerotic lesions. Our data indicated that the knockout of Xcl1 led to a mild yet significant reduction of cDC1 cells in the aorta. However, strikingly, when compared to the *Apoe$^{-/-}$* control mice, *Xcl1$^{-/-}$ Apoe$^{-/-}$* mice exhibited a markedly decreased severity of the atherosclerotic lesions, to an extent comparable to that observed in cDC1-depleted *Xcr1$^{Cre-Gfp}$ Rosa26$^{LSL-DTA}$ Apoe$^{-/-}$* mice.

Therefore, our study, which employed genetic models to deplete Xcr1$^+$ cDC1 in vivo in hyperlipidemic mice, has unequivocally determined the essential role of this specific cell type in the development of atherosclerosis. Besides their well-established role in cross-presentation to CD8$^+$ T cells,

recent studies have shown Xcr1+ cDC1 cells also engage with CD4+ T cells to augment CD8+ T cell responses (*Wohn et al., 2020*; *Eickhoff et al., 2015*). Our data showing Xcr1+ cDC1-dependent CD4 and CD8 T cell activation in atherosclerotic plaques is in line with the new paradigm for cDC1s as a platform for both CD4 and CD8 T cell responses in the novel context of atherosclerosis.

Most importantly, we have successfully demonstrated that targeting the chemokine Xcl1, the ligand of Xcr1, can effectively inhibit atherosclerosis. These novel findings hold the potential to forge a novel trajectory for therapeutic interventions within the field of atherosclerosis, presenting promising prospects for better understanding and treatment of this disorder.

## Results

### Progressive accumulation of Xcr1+ cDC1 cells in advanced atherosclerotic lesions in humans and mice

Dendritic cells are present in healthy arteries and accumulate within atherosclerotic lesions, participating in various pathogenic and protective mechanisms during atherogenesis (*Weber et al., 2008*; *Worbs et al., 2017*). However, the intra-plaque dynamics of cDC1 and their contribution to atherosclerosis remain undetermined. Using the archived specimens of paraffin-embedded human

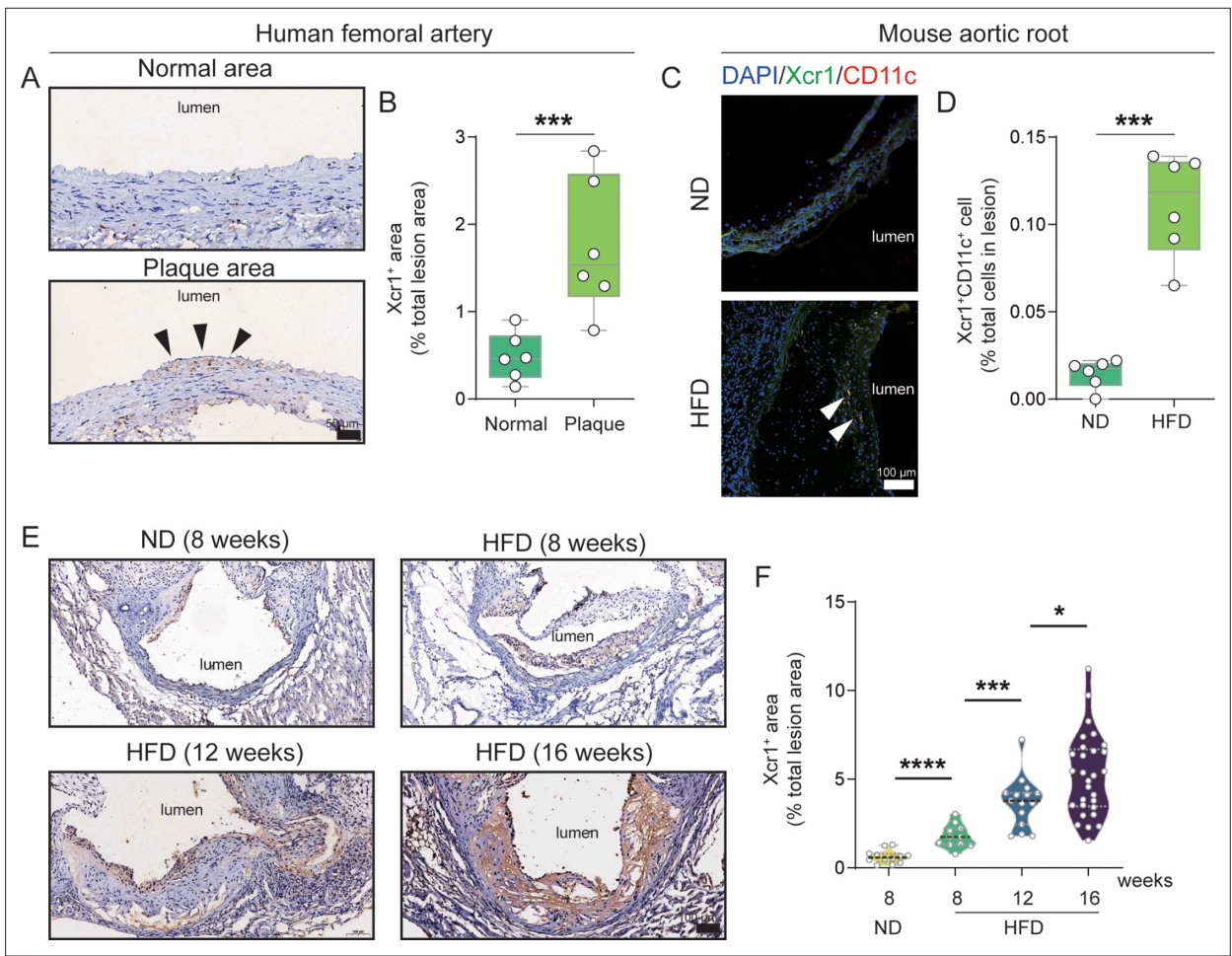

**Figure 1.** Progressive accumulation of Xcr1+ cDC1 cells in human and mouse advanced atherosclerotic lesions. (**A**) Representative images of immunohistochemical staining demonstrate Xcr1 expression in normal areas and plaque areas of the femoral artery from three clinical samples. (**B**) Quantitative analysis of the Xcr1-positive area in normal area and plaque regions. (**C**) Representative immunofluorescence images depict DAPI (blue), Xcr1 (green), and CD11c (red) within the lesions of aortic root of *Apoe*−/− mice fed with 16 week normal diet (ND) or high-fat diet (HFD). (**D**) Quantitative analysis of Xcr1 and CD11c double positive area in lesion area (n=6). Scale bars, 100 μm. (**E**) Representative images depicting immunohistochemical staining of Xcr1 in lesions of the aortic root of *Apoe*−/− mice fed a HFD for varying durations. (**F**) Quantitative analysis of the Xcr1-positive area in the lesion regions. (8 week HFD, n=11; 12 week HFD, n=15; 16 week HFD, n=29). Data represent as mean ± SEM. *p<0.05. ***p<0.001.****p<0.0001.

atherosclerotic arteries from our previous study (*Li et al., 2020*), we examined the Xcr1$^+$ dendritic cells in human atherosclerotic plaques. As shown in *Figure 1A and B*, Xcr1$^+$ dendritic cells were significantly more abundant in the plaque area of the human femoral artery compared to that in the plaque-free area. Similarly, the immunofluorescence staining results using specific antibodies for Xcr1 and CD11c revealed that Xcr1$^+$CD11c$^+$ dendritic cells accumulated in the plaques of the aortic root in *Apoe*$^{-/-}$ mice that were placed on a HFD for 16 weeks (*Figure 1C and D*). Sample preparations entailed sectioning and appropriate fixation of the aortic root tissues, which were described in greater detail in the Methods and Materials section. To further elucidate the dynamics of Xcr1$^+$ cDC1 cells within the plaque during the progression of atherosclerosis, we carried out immunohistochemical analyses of these cells in the lesioned aortic root (*Figure 1E*). The area positive for Xcr1 expression was quantified at 8 week on a normal diet (ND), 8 week on HFD, 12 week on HFD and 16 week on HFD, respectively. Quantitative analysis was conducted using image processing software, revealing a continuous accumulation of Xcr1$^+$ cDC1 cells in the lesioned area as the disease progressed (*Figure 1F*). Therefore, our experiments provided data indicating the enrichment of Xcr1$^+$ cDC1 cells in aortic lesions in both humans and mice, and demonstrated that the accumulation of Xcr1$^+$ cDC1 cells was significantly greater during the progression of atherosclerosis.

## *Xcr1*$^{Cre-Gfp}$ is selectively expressed in cDC1

To understand the role of Xcr1$^+$ cDC1 in regulating the development of atherosclerosis, we established the *Xcr1*$^{Cre-Gfp}$ *Apoe*$^{-/-}$ mice in which the Cre recombinase and eGFP was co-expressed under the control of the endogenous Xcr1 promoter following the start codon (*Figure 2—figure supplement 1*). This new Cre line differs from a previous design in which Cre recombinase was expressed via IRES linkage at the 3′ UTR, and leaked expression of Cre was frequently observed (*Wohn et al., 2020*; *Mattiuz et al., 2018*). To verify the fidelity of Cre expression under hyperlipidemic conditions, we used hyperlipidemic *Apoe*$^{-/-}$ mice as recipient mice and used bone marrow cells from *Xcr1*$^{Cre-Gfp}$ *Rosa26*$^{LSL-tdRFP}$ mice as donors for the bone marrow transplantation (*Figure 2A*). In *Xcr1*$^{Cre-Gfp}$ *Rosa26*$^{LSL-tdRFP}$ mice, the presence of Cre protein would turn on the expression of tdRFP as a reporter for the activity of Cre recombinase (*Luche et al., 2007*). Remarkably, we found that the expression of tdRFP was exclusively present in Xcr1$^+$ cDC1 when mice were fed a HFD for 16 weeks in the bone marrow transplanted mice, and the Cre recombinase activity was not detectable in non-cDC1 cells (*Figure 2B and C*).

To determine the specificity of GFP expression originating from *Xcr1*$^{Cre-Gfp}$ genetic modification, we examined the GFP levels in three populations of DCs, namely pDC, cDC1, and cDC2 cells, from the spleen of *Xcr1*$^{Cre-Gfp}$ *Apoe*$^{-/-}$ mice that were fed a HFD for 16 weeks. As expected, we found that the GFP expression was stringently restricted to cDC1 cells (*Figure 2D and E*). In parallel, we compared the GFP expression in Xcr1$^+$ cDC1-depleted mice that were fed a HFD for 16 weeks. Such cDC1-depleted *Xcr1*$^{Cre-Gfp}$ *Rosa26*$^{LSL-DTA}$ *Apoe*$^{-/-}$ mice were obtained by crossing *Xcr1*$^{Cre-Gfp}$ *Apoe*$^{-/-}$ mice with *Rosa26*$^{LSL-DTA}$ *Apoe*$^{-/-}$ mice. Strikingly, we observed a complete loss of GFP-positive cells and Xcr1$^+$ cDC1 cells in Xcr1$^+$ cDC1-depleted mice (*Figure 2D and F*). We also examined DCs, including pDCs, cDC1, and cDC2, in the lymph nodes and spleens of *Xcr1*$^{Cre-Gfp}$ *Rosa26*$^{LSL-DTA}$ *Apoe*$^{-/-}$ and *Apoe*$^{-/-}$ mice maintained on a ND for 7 weeks. The flow cytometric analyses also confirmed the absence of Xcr1$^+$ cDC1 population in *Xcr1*$^{Cre-Gfp}$ *Rosa26*$^{LSL-DTA}$ *Apoe*$^{-/-}$ mice (*Figure 2—figure supplement 2*). Collectively, these data suggest that Cre recombinase activity was selectively confined to cDC1 cells, and no leakiness was observed in lineages other than the Xcr1$^+$ cDC1 population. We also demonstrated that *Xcr1*$^{Cre-Gfp}$ *Rosa26*$^{LSL-DTA}$ *Apoe*$^{-/-}$ mice constitute a desirable model for exploring the role of cDC1 in atherosclerosis.

## The specific depletion of Xcr1$^+$ cDC1 cells significantly alleviates atherosclerosis without influencing the lipid profile and the number of intra-plaque macrophages

To investigate the role of cDC1 cells in atherosclerosis, both *Xcr1*$^{Cre-Gfp}$ *Rosa26*$^{LSL-DTA}$ *Apoe*$^{-/-}$ and the control *Apoe*$^{-/-}$ mice were subjected to a HFD for 16 weeks. The Oil Red O (ORO) staining results demonstrated that the size of the lesion in both descending aorta and aortic root was significantly smaller in *Xcr1*$^{Cre-Gfp}$ *Rosa26*$^{LSL-DTA}$ *Apoe*$^{-/-}$ mice compared with Apoe$^{-/-}$ control mice (12.82% *vs* 6.01% in descending aorta; 39.25% *vs* 33.57% in ORO-stained aortic root) (*Figure 3A–D*). Given

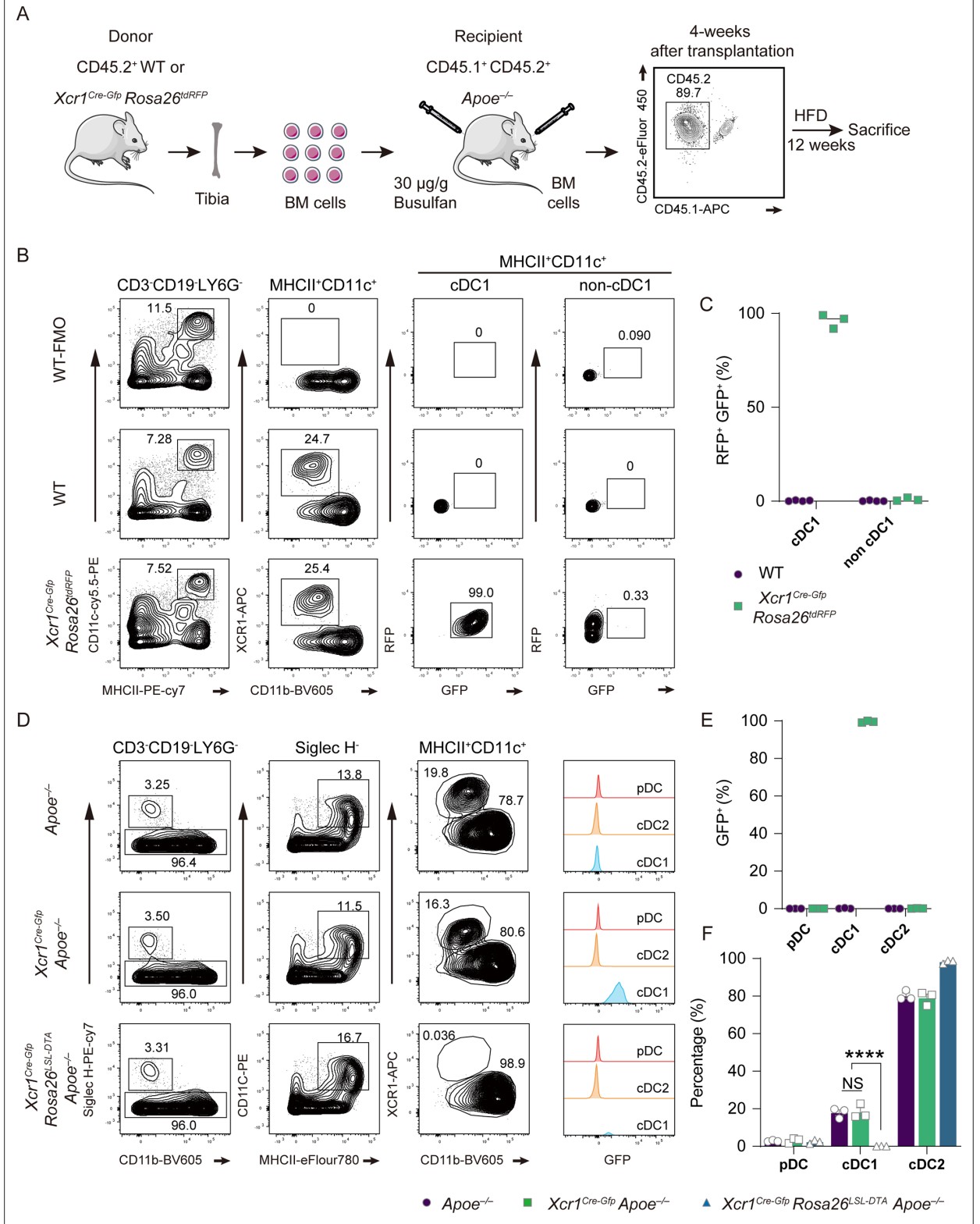

**Figure 2.** Selective expression of *Xcr1^Cre-Gfp* in cDC1 cells. (**A**) Diagram illustrating the bone marrow transfer process. (**B**) and (**C**) Representative flow cytometric analysis and corresponding quantification of GFP⁺RFP⁺ in cDC1 and non-cDC1 populations within the cDC1 and non-cDC1 populations of the spleen. These data were obtained from *Apoe^−/−* mice that had received bone marrow transplants from either wild-type (WT) or *Xcr1^Cre-Gfp Rosa26^LSL-RFP* donors and were maintained on a HFD for 16 weeks (WT, n=4; *Xcr1^Cre-Gfp Rosa26^LSL-RFP*, n=3). (**D**) and (**E**) Representative flow cytometric analysis and quantification of the percentage of GFP⁺ cells among pDC, cDC1, and cDC2 cells populations in the spleen. (**F**) Quantification of the percentage

*Figure 2 continued on next page*

*Figure 2 continued*

of pDC, cDC1, and cDC2 cells in spleen. (D-F, *Apoe*⁻/⁻, *Xcr1^Cre-Gfp Apoe*⁻/⁻, and *Xcr1^Cre-Gfp Rosa26^LSL-DTA Apoe*⁻/⁻ mice fed with a high-fat diet (HFD) for 16 weeks, n=3). Data represent as mean ± SEM. **** p<0.0001; NS, non-significant.

The online version of this article includes the following figure supplement(s) for figure 2:

**Figure supplement 1.** Schematic diagram illustrating the knock-in of the 5'HA-iCre-P2A-EGFP-P2A-3'HA vector into the mouse Xcr1 locus.

**Figure supplement 2.** Flow cytometric analysis of cDC1 cells in the lymph nodes and spleens of *Apoe*⁻/⁻ mice and *Xcr1^Cre-Gfp Rosa26^LSL-DTA Apoe*⁻/⁻ mice fed a 7 weeks normal diet (ND).

that macrophages play a pivotal role in the development and progression of atherosclerotic lesions, particularly within the necrotic core (*De Meyer et al., 2024*; *Yu et al., 2017*), we next investigated macrophage involvement in the aorta using IHC and flow cytometry. Interestingly, no significant differences were observed in the necrotic core area of the aortic root between *Xcr1^Cre-Gfp Rosa26^LSL-DTA Apoe*⁻/⁻ mice and *Apoe*⁻/⁻ control mice (*Figure 3E and F*). Similarly, subsequent flow cytometry analyses and IHC results revealed comparable proportions of F4/80⁺ CD11b⁺ macrophages in the aorta, as well as similar percentages of CD68-positive macrophages in the aortic lesions between the two groups of mice (*Figure 3G–J*). Furthermore, since dyslipidemia is a key characteristic of HFD-treated *Apoe*⁻/⁻ mice (*Piedrahita et al., 1992*; *Plump et al., 1992*), it is critical to assess body weight and lipid profiles in both groups. Our results indicated no significant differences in body weight or levels of total cholesterol (TC), triglycerides (TG), low-density lipoprotein (LDL), and high-density lipoprotein (HDL) between the two groups (*Figure 3—figure supplement 1A–B*). Additionally, ORO staining of the liver revealed no significant differences either (*Figure 3—figure supplement 1C–D*). Taken together, these results suggested that specific depletion of cDC1 attenuated atherogenesis without affecting lipid status or macrophagic accumulation in aortic lesions.

## Specific depletion of cDC1 attenuates the development of atherosclerosis by modulating the activation of T cell in *Apoe*⁻/⁻ mice

T cells are among the critical drivers of the pathogenesis of atherosclerosis, and therefore, the triggers of T cell activation in aortic plaques are of great interest for controlling disease progression (*Hansson and Hermansson, 2011*; *Fernandez et al., 2019*; *Afek et al., 2004*). We initially analyzed the cDC1 cells through flow cytometry in both Xcr1⁺ cDC1-depleted mice and control mice. In the *Xcr1^Cre-Gfp Rosa26^LSL-DTA Apoe*⁻/⁻ mice that were fed a HFD for 16 weeks, we observed a total absence of Xcr1⁺ cDC1 in the aorta, lymph nodes, and spleen (*Figure 4A-C*, *Figure 4—figure supplement 1*). Subsequently, we assessed T cell phenotype in the two groups of mice. While neither the frequencies nor absolute counts of aortic CD4⁺ and CD8⁺ T cells differed significantly between the two groups of mice (*Figure 4D–F*), CD69 frequency and CD44 MFI (Mean Fluorescence Intensity), the T cell activation markers, were significantly reduced in both CD4⁺ and CD8⁺ T cells from Xcr1⁺ cDC1 depleted mice compared to controls (*Figure 4G and H*). Interestingly, such decreased frequencies of CD69⁺ T cells were only observed in aortic lesions but not in lymphoid organs such as lymph nodes and the spleen in the Xcr1⁺ cDC1 depleted mice (*Figure 4—figure supplement 2A–B and D-E*). We further analyzed the absolute numbers of CD8⁺ and CD4⁺ T cells in the spleen and lymph nodes and found that the absolute numbers of CD8⁺ and CD4⁺ T cells were markedly reduced in the spleen but not in the lymph nodes in *Xcr1^Cre-Gfp Rosa26^LSL-DTA Apoe*⁻/⁻ mice compared to those in *Apoe*⁻/⁻ mice (*Figure 4—figure supplement 2C and F*). Absence of cDC1 also resulted in reduced frequencies of CD8⁺ T cells in both spleen and lymph nodes in hyperlipidemic mice. We also analyzed T cell populations in the lymph nodes and spleen in two groups of mice that were fed a ND and found that the frequencies of CD8⁺ T cells were significantly decreased in the lymph nodes of Xcr1⁺ cDC1 depleted mice (*Figure 4—figure supplement 3A–D*). More importantly, the frequencies of CD44^high CD62L^low activated memory cells within the CD8⁺ subsets in both the spleen and lymph nodes were significantly lower than those in the *Apoe*⁻/⁻ control mice (*Figure 4—figure supplement 3A–D*). However, in the CD4⁺ subsets, such frequencies of CD44^high CD62L^low activated cells were comparable in the lymph nodes but marginally lower in the spleen (*Figure 4—figure supplement 3A–D*). It is important to note that the decreased absolute number of CD8⁺ T cells in the spleen in Xcr1⁺ cDC1-depleted mice was only observed in HFD groups but not in ND groups (*Figure 4—figure supplements 2 and 3*). Our data indicate that in aortic plaques, Xcr1⁺ cDC1 cells are implicated in activating CD4⁺ and CD8⁺ T cells, and the loss of the cDC1

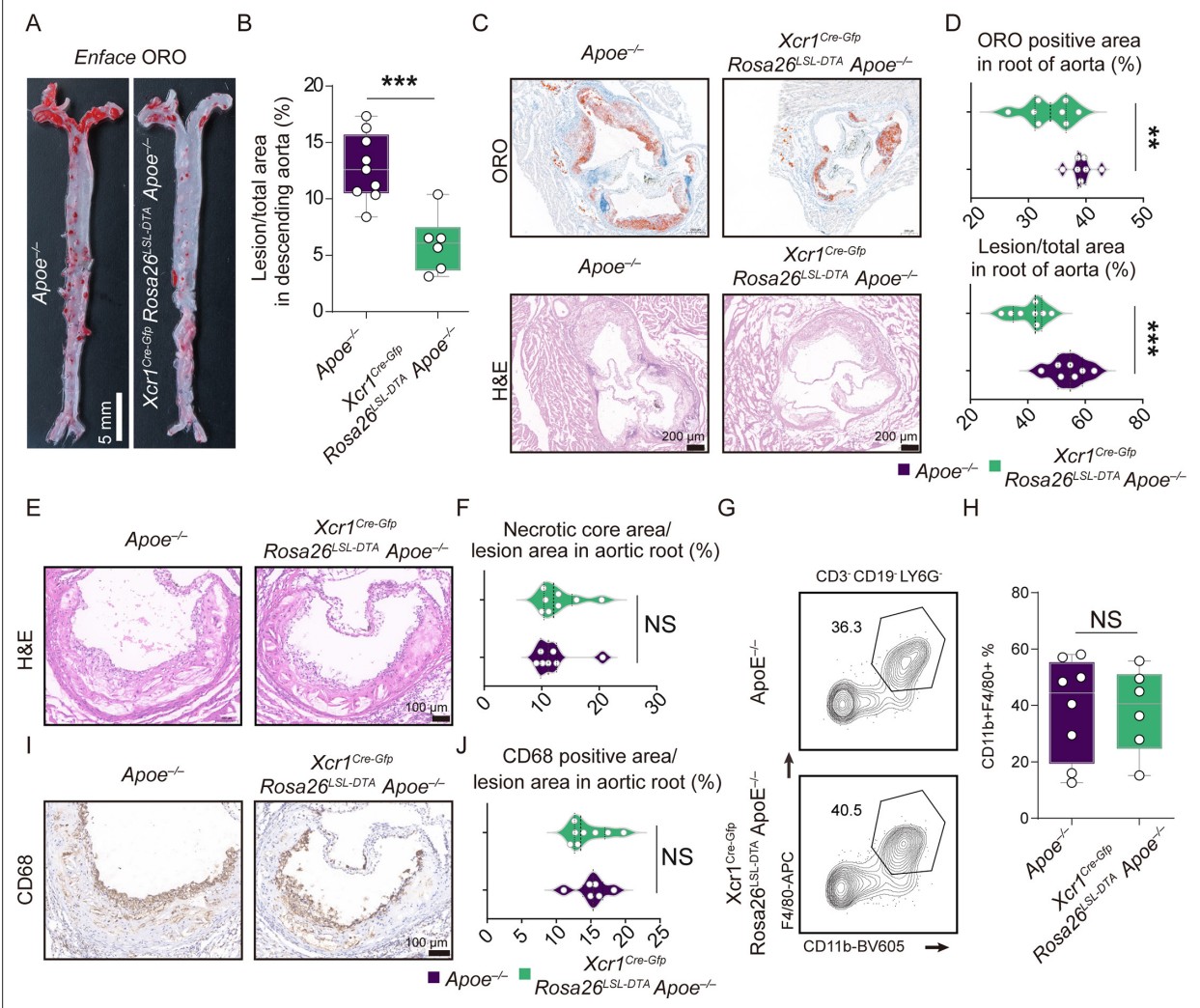

**Figure 3.** Specific depletion of Xcr1[+] cDC1 cells in *Apoe*[−/−] mice reduces atherosclerosis progression with no effect on macrophages. (**A**) Oil Red O (ORO) staining of the descending aortas. Scale bar, 5 mm. (**B**) Quantification of the lesion area in the descending aorta (*Apoe*[−/−] mice, n=9; *Xcr1*[Cre-Gfp] *Rosa26*[LSL-DTA] *Apoe*[−/−] mice, n=6). (**C**) ORO and H&E staining of the aortic roots of representative mice from each group. (**D**), Quantification of lesion area in the aortic root. (*Apoe*[−/−] mice, n=7; *Xcr1*[Cre-Gfp] *Rosa26*[LSL-DTA] *Apoe*[−/−] mice, n=8). Scale bar, 200 μm. (**E**) and (**F**) H&E staining and quantification of necrotic core area in the lesions of aortic root. (*Apoe*[−/−] mice, n=7; *Xcr1*[Cre-Gfp] *Rosa26*[LSL-DTA] *Apoe*[−/−] mice, n=8). Scale bar, 200 μm. (**G** and **H**) Representative flow cytometric analysis and quantification of macrophages in the aorta. (*Apoe*[−/−] mice, n=8; *Xcr1*[Cre-Gfp] *Rosa26*[LSL-DTA] *Apoe*[−/−] mice, n=6).(**I** and **J**), Representative immunohistochemical staining images of CD68 and quantification of CD68-positive area in the lesions of the aortic root. (*Apoe*[−/−] mice, n=6; *Xcr1*[Cre-Gfp] *Rosa26*[LSL-DTA] *Apoe*[−/−] mice, n=8). Scale bar, 100 μm. *Apoe*[−/−] and *Xcr1*[Cre-Gfp] *Rosa26*[LSL-DTA] *Apoe*[−/−] mice were fed on a 16 week high-fat diet (HFD) to develop atherosclerosis. Data represent as mean ± SEM. NS, non-significant. **p<0.01. ***p<0.001.

The online version of this article includes the following figure supplement(s) for figure 3:

**Figure supplement 1.** Lipid profile, body weight and Oil Red O (ORO) staining of the liver in *Apoe*[−/−] mice and *Xcr1*[Cre-Gfp] *Rosa26*[LSL-DTA] *Apoe*[−/−] mice fed a 16 week high-fat diet (HFD).

population led to a decreased absolute count of CD8[+] T cells in the lymph nodes of HFD-treated mice. Whereas in mice treated with a ND, the loss of the cDC1 population resulted in reduced frequencies of activated memory T cells but did not affect the absolute counts of T cells. Attenuated T cell activation might account for the mitigated atherosclerosis in Xcr1[+] cDC1-depleted mice.

## Characterization of aortic Xcr1[+] cDC1 cells in atherosclerotic lesions

Although the majority of cDC1 cells derive from bone marrow, other mechanisms, such as local proliferation of existing DCs, can also contribute to the cDC1 population in inflammatory diseases (*Cancel et al., 2019*; *Li et al., 2021*). It remains unclear whether cDC1 cells in lymphoid organs, such as the

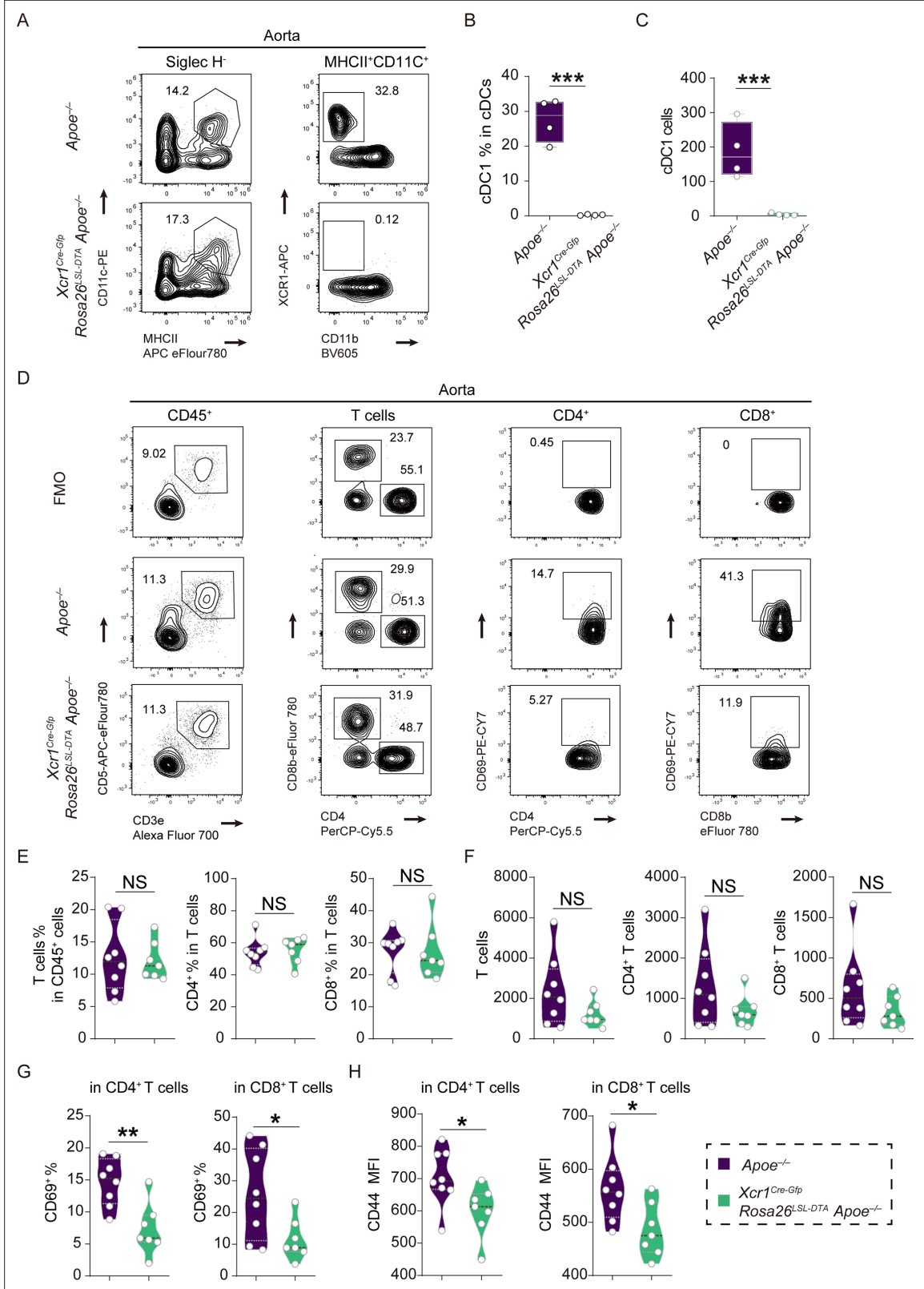

**Figure 4.** Specific depletion of Xcr1+ cDC1 cells in *Apoe*−/− mice decreases T cell activation within aorta. (**A**) Representative flow cytometric analysis of cDC1 cells in aortas. (**B** and **C**) Quantification of the frequencies and absolute counts of cDC1 cells in aorta (n=4). (**D**) Representative flow cytometric analysis T cells in aortas. (**E** and **F**) Quantification of the frequencies and absolute counts of total T, CD4+, and CD8+ T cells in aortas. (**G**) Quantification of the frequencies of CD4+CD69+ and CD8+ CD69+ in aortas. (**H**) Quantification of the MFI of CD44 in CD4+ and CD8+ T cells in aortas. (D-G, *Apoe*−/−

*Figure 4 continued on next page*

*Figure 4 continued*

mice, n=8, *Xcr1*<sup>Cre-Gfp</sup> *Rosa26*<sup>LSL-DTA</sup> *Apoe*<sup>–/–</sup> mice, n=7, fed on a high-fat diet HFD for 16 weeks). Data represent as mean ± SEM. *p<0.05, **p<0.01, ***p<0.001, NS, non-significant.

The online version of this article includes the following figure supplement(s) for figure 4:

**Figure supplement 1.** Flow cytometric analysis of cDC1 cells in the lymph nodes and spleens of *Apoe*<sup>–/–</sup> and *Xcr1*<sup>Cre-Gfp</sup> *Rosa26*<sup>LSL-DTA</sup> *Apoe*<sup>–/–</sup> mice fed a 16 week high-fat diet (HFD).

**Figure supplement 2.** Flow cytometric analysis of T cells in the lymph nodes and spleens of *Apoe*<sup>–/–</sup> mice and *Xcr1*<sup>Cre-Gfp</sup> *Rosa26*<sup>LSL-DTA</sup> *Apoe*<sup>–/–</sup> mice fed a 16 week high-fat diet (HFD).

**Figure supplement 3.** T cell analysis in the lymph nodes and spleens of *Apoe*<sup>–/–</sup> mice and *Xcr1*<sup>Cre-Gfp</sup> *Rosa26*<sup>LSL-DTA</sup> *Apoe*<sup>–/–</sup> mice fed a 7 week normal diet (ND).

spleen and lymph nodes, represent a different status compared to cDC1 cells in atherosclerotic lesions. To elucidate the source and status of Xcr1$^+$ cDC1 in atherosclerotic lesions, we performed bone marrow transplantation and single-cell RNA sequencing (sc-RNA seq) of cDC1 cells from aortic plaques and lymphoid organs. We conducted bone marrow transplantation using *Xcr1*<sup>Cre-Gfp</sup> *Rosa26*<sup>LSL-DTA</sup> or WT mice that carry the CD45.2 congenic markers as donors, and CD45.1$^+$ CD45.2$^+$*Apoe*<sup>–/–</sup> mice as recipients (*Figure 5—figure supplement 1*). Our data showed that body weight, lipid profile, and hepatic lipid accumulation were comparable between the two groups of bone marrow transplanted mice after being fed with a HFD for 16 weeks (*Figure 5—figure supplement 1*). Interestingly, flow cytometric analysis of the aorta displayed almost complete depletion of cDC1 cells in *Apoe*<sup>–/–</sup> recipient mice transplanted with *Xcr1*<sup>Cre-Gfp</sup> *Rosa26*<sup>LSL-DTA</sup> bone marrow mice (*Figure 5—figure supplement 1*). Strikingly we did not find observe significant changes in pDCs and CD64$^+$ F4/80$^+$ aortic macrophages (*Figure 5—figure supplement 1*). Therefore, our bone marrow transplantation experiments further confirmed fidelity of the Cre recombinase expression in cDC1 cells in mice.

Next, we carried out sc-RNA seq to characterize Xcr1$^+$ cDC1 cells from multiple organs of hyperlipidemic mice. Due to the scarcity of Xcr1$^+$ cDC1 cells that can be sorted by FACS and processed for sc-RNA seq, we pooled FACS-sorted Xcr1$^+$ cDC1 cells from three organs of hyperlipidemic *Apoe*<sup>–/–</sup> mice fed a HFD for 20 weeks, including the aorta, spleen, and lymph nodes, and Xcr1$^+$ cDC1 cells from aorta and splenocytes of *Apoe*<sup>–/–</sup> mice fed a ND. Libraries for sc-RNA seq were prepared using barcode oligos conjugated to anti-CD45 antibodies for surface staining so that samples from each type of organ could be compared in sequential analyses. Therefore, we had a total of five samples, and our focus was to understand the differences between Xcr1$^+$ cDC1 cells from three different organs under hyperlipidemic conditions. The cells were sorted by gating Xcr1-positive cells out of the MHCI-I$^+$CD11c$^+$ conventional dendritic cells (*Figure 5—figure supplement 2*). First, we obtained the UMAP plot of 10 clusters of cDC1 cells using the Seurat 4.0 package (*Figure 5A*). In this analysis, 3830 quality-controlled Xcr1$^+$ cDC1 cells were from three types of organs in hyperlipidemic mice, and an additional 2081 cells were from cDC1 cells sorted from the spleen and aorta of *Apoe*<sup>–/–</sup> mice fed on a ND. The clusters were differentiated according to top 10 differentially expressed genes (*Supplementary file 1*), and the heatmap shows marker genes and heterogeneity among the 10 clusters of cDC1 cells (*Figure 5B*). The purpose of this experiment was to determine whether cDC1 cells from different organs possess unique features and whether cDC1 cells from aortic lesions can be characterized by the expression of marker genes. In hyperlipidemic mice, cDC1 cells from the spleen and lymph nodes differed quite dramatically in Cluster 2 (*Stk17b* and *Gpr171* highly expressing cells), where 28.9% of cDC1 cells from the spleen versus 1.0% of cDC1 cells from the lymph nodes were located. Conversely, 3.15% of cDC1 cells from the spleen versus 32.4% of cDC1 cells from the lymph nodes fell in Cluster 3 (*Ccl5* and *Epsti1* highly expressing cells), and 4.6% of cDC1 cells from the spleen versus 25.9% of cDC1 cells from the lymph nodes fell in Cluster 4 (*Slfn5* and *Ifi44* highly expressing cells). Despite the fact that we had only transcriptomic data for 68 Xcr1$^+$ cDC1 cells from aortic lesions, we were surprised to find that such aortic cDC1 mainly fell into Clusters 1, 4, and 5, comprising 80.8% of the cells analyzed (*Figure 5C*). In Cluster 1 (*Hspa1b* and *Dnajb1* highly expressing cells), where aortic cDC1 is dominantly enriched, the cDC1 cells sorted from lymph nodes were essentially absent, while in Cluster 4, aortic cDC1 cells were highly enriched and splenic cDC1 was obviously lower in percentage (*Figure 5C*). Cluster 7 (*Fabp5* and *S100a4* highly expressing cells) is exclusively comprised of cDC1 cells sourced from lymph nodes, with a striking absence of cDC1 cells from the spleen and aorta (*Figure 5C*). To

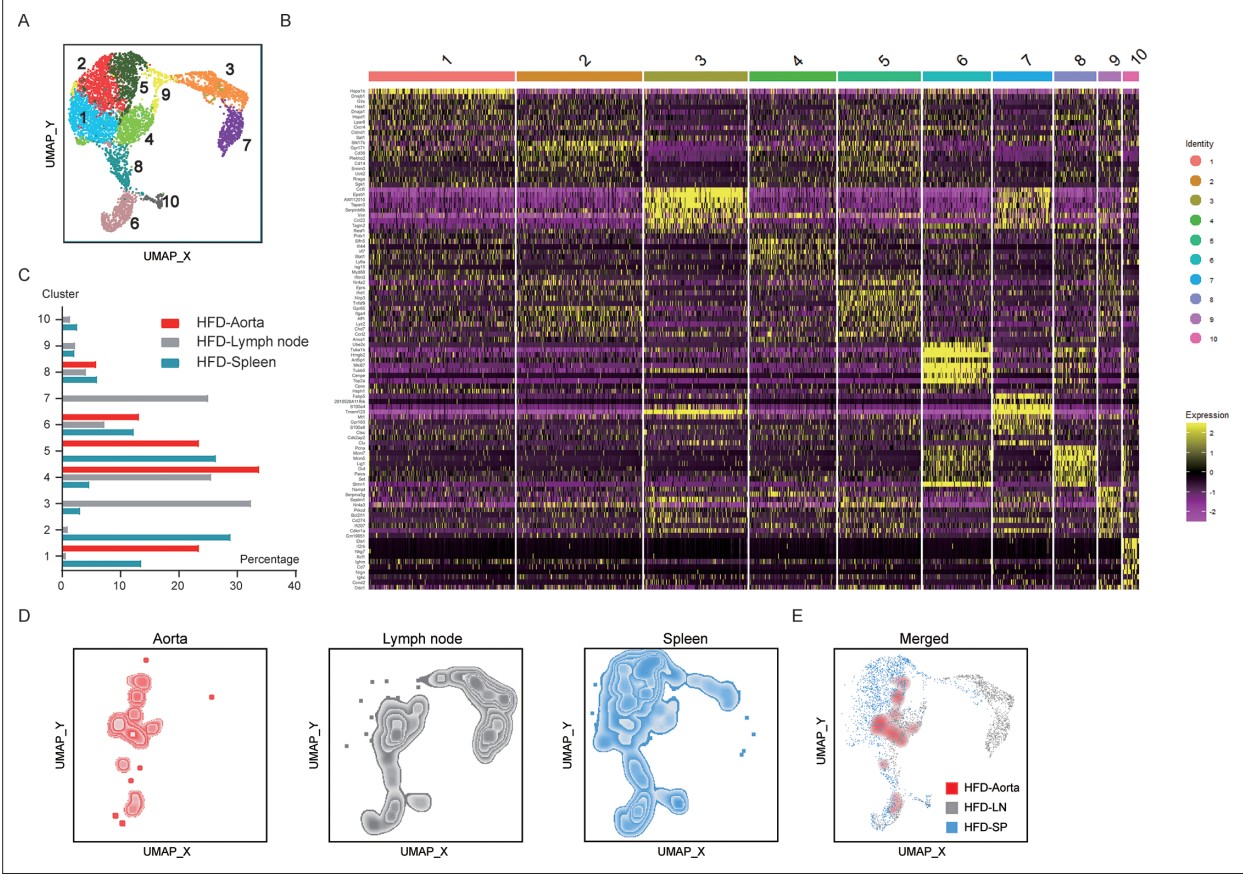

**Figure 5.** Sc-RNA sequencing analysis of the aorta, lymph nodes and spleen in *Apoe⁻/⁻* mice fed with a 20 week high-fat diet (HFD). (**A**) UMAP plot delineates 10 annotated cell types of Xcr1⁺ cDC1 cells from aorta, lymph node and spleen in *Apoe⁻/⁻* mice maintained on a HFD or normal diet (ND) for 20 weeks. (**B**) Heatmap statistic map displays the top ten up-regulated genes across various clusters. (**C**) The proportion of 10 clusters among groups. (**D** and **E**) UMAP plot representing Xcr1⁺ cDC1 cells from aorta, lymph nodes, spleen, and a merged plot in *Apoe⁻/⁻* mice maintained on a HFD for 20 weeks.

The online version of this article includes the following figure supplement(s) for figure 5:

**Figure supplement 1.** Depletion of Xcr1+ cDC1 cells of bone marrow reduces cDC1 cells without affecting macrophages in atherosclerotic lesion of Apoe⁻/⁻ mice.

**Figure supplement 2.** The single-cell RNA sequencing process.

**Figure supplement 3.** Sc-RNA sequencing analysis of the aorta, lymph nodes and spleen in Apoe⁻/⁻ mice.

**Figure supplement 4.** Single-cell RNA sequencing of splenic cDC1s from both the high-fat diet (HFD) and normal diet (ND) groups.

better visualize the enrichment of aortic Xcr1⁺ cDC1 cells distributed in a few clusters, we separately compared the three samples from the spleen, lymph nodes, and aorta of hyperlipidemic mice using density plots. Interestingly, the Xcr1⁺ cDC1 cells from the spleen and lymph nodes displayed an obvious difference in their distribution, and aortic Xcr1⁺ cDC1 cells appeared more enriched in a few clusters (*Figure 5D–E*). Therefore, our data suggest that Xcr1⁺ cDC1 cells from the spleen and lymph nodes can differ markedly in heterogeneity, and aortic Xcr1⁺ cDC1 cells are enriched in a few clusters that express marker genes, *Ccr2, Sept3,* and *Cldnd1* (*Supplementary file 2*). Using the scRNA seq data of Xcr1⁺ cDC1 sorted from hyperlipidemic mice, we annotated the 10 populations as shown in *Figure 5—figure supplement 3A*, following the methodology from a previous study (*Bosteels et al., 2023*). *Ccr7⁺* mature cDC1s (Clusters 3, 7, and 9) and *Ccr7*- immature cDC1s (remaining clusters) were identified across cDC1 cells sorted from aorta, spleen and lymph nodes (+). Further stratification based on marker genes reveals that Cluster 10 is the pre-cDC1, with high expression level of *Sell* (*Cd62l*) and low expression level of *Cd8a* (*Figure 5—figure supplement 3C*). Clusters 6 and 8 are the proliferating cDC1s, which express high level of cell cycling genes *Stmn1* and *Top2a*. Cluster 1 and 4 are early immature cDC1s, and cluster 2 and 5 are late immature cDC1s, according to the expression

pattern of *Itgae*, *Nr4a2*. Cluster 9 cells are early mature cDC1s, with elevated expression of *Cxcl9* and *Cxcl10*. Cluster 3 and 7 as late mature cDC1s, characterized by the expression of *Cd63* and *Fscn1*. As shown in *Figure 5*, the 10 populations displayed a major difference of aortic cDC1 cells that lack in pre-cDC1s (cluster 10) and mature cells (cluster 3, 7, and 9). Interestingly, in hyperlipidemic mice splenic cDC1 possess only Cluster 3 as the late mature cells while the lymph node cDC1 cells have two late mature populations, namely Cluster 3 and Cluster 7. In further analysis, we also compared splenic cDC1 cells from HFD mice to those from ND mice. As shown in , HFD appears to impact early immature cDC1-1 cells (Cluster 1) and increases the abundance of late immature cDC1 cells (Cluster 2 and 5), regardless of the fact that all 10 populations are present in two origins of samples. We also found that *Tnfaip3* and *Serinc3* are among the most upregulated genes, while *Apol7c* and *Tifab* are downregulated in splenic cDC1 cells sorted from HFD mice.

Together, these data implied that Xcr1$^+$ cDC1 in the aorta originate from bone marrow, and Xcr1$^+$ cDC1 cells exhibit strong heterogeneity among aorta, spleen, and lymph nodes.

## Knockout of *Xcl1* attenuates the development of atherosclerosis in *Apoe*$^{-/-}$ mice

Xcl1 is the major chemokine involved in cDC1 recruitment (*Böttcher et al., 2018*; *Dorner et al., 2009*), and plays a critical role in immune response with diverse physical and pathological implication through its interaction with the sole receptor, Xcr1 (*Xu et al., 2019*; *Yoshida et al., 1999*). We found the presence of Xcl1 in the lesion area in both human and mouse atherosclerotic lesions (data not shown). To elucidate the causal relationship between Xcl1 and Xcr1$^+$ cDC1 in the development of atherosclerosis, and more importantly, to experimentally verify Xcl1 as a potential therapeutic target for atherosclerosis treatment, we established the *Xcl1* and *Apoe* double knockout mice. After being fed with a HFD for 19 weeks, both *Xcl1*$^{-/-}$ *Apoe*$^{-/-}$ and *Apoe*$^{-/-}$ mice were sacrificed to assess the severity of atherosclerosis. As shown in *Figure 6A–D*, the size of lesion at the descending aorta and aortic root was significantly smaller in *Xcl1*$^{-/-}$ *Apoe*$^{-/-}$ compared to that in *Apoe*$^{-/-}$ mice. Importantly, ORO staining of the liver, body weight and blood lipid profiles revealed no significant differences between the two groups (*Figure 6E–H*). Moreover, the IHC experiment result shows the percentage of CD68 positive macrophages in aortic lesion was similar between two groups (*Figure 6I–J*).

Previous study identified that Xcl1 serves as a potent chemokine, selectively acting on cDC1 cells through its exclusive receptor, Xcr1 (*Dorner et al., 2009*). Consequently, we examined the DCs and the activation of T cells in different organs, including aorta, lymph node, and spleen. The flow cytometric results illustrated that both frequencies and absolute counts of Xcr1$^+$ cDC1 cells in the aorta were significantly reduced, but cDCs and cDC2 cells from *Xcl1*$^{-/-}$ *Apoe*$^{-/-}$ were comparable with that from *Apoe*$^{-/-}$ (*Figure 7A–C*). Moreover, in both lymph node and spleen, the absolute numbers of pDC, cDC1, and cDC2 from *Xcl1*$^{-/-}$ *Apoe*$^{-/-}$ were comparable with that from *Apoe*$^{-/-}$ (*Figure 7—figure supplement 1*). Crucially, CD69$^+$ frequency and CD44 MFI remained comparable in both aortic CD4$^+$ and CD8$^+$ T cells between two groups (*Figure 7D–F*). However, aortic CD8$^+$ T cells exhibited reduced frequency and absolute count, while CD4$^+$ T cells showed increased frequency but unchanged counts in *Xcl1*$^{-/-}$ *Apoe*$^{-/-}$ mouse versus controls (*Figure 7G and H*). Furthermore, the absolute numbers of CD4$^+$, CD8$^+$, CD4$^+$ CD69$^+$, and CD8$^+$ CD69$^+$ T cells in both lymph node and spleen from *Xcl1*$^{-/-}$ *Apoe*$^{-/-}$ mouse were comparable with that from *Apoe*$^{-/-}$ mouse (*Figure 7—figure supplement 2*).

Collectively, these data suggest that Xcl1 promotes atherosclerosis by recruiting Xcr1$^+$ cDC1 cells, and facilitating CD8$^+$ T cell accumulation in lesions. More importantly, in vivo data demonstrated the promising potential of targeting Xcl1 to reduce the inflammatory response in atherosclerotic lesions. Our work paves the way to discover novel solutions for the treatment of atherosclerosis by targeting Xcl1 and Xcr1$^+$ cDC1 cells.

## Discussion

Atherosclerosis is a complex disorder, and understanding the role of the immune system, particularly dendritic cells (DCs), is of paramount importance (*Tall and Bornfeldt, 2023*; *Vallejo et al., 2021*). The marked diversity of DC subsets and their distinct, and at times contradictory functions pose challenges in clarifying the role of each specific type of DC in the context of atherosclerosis (*Guo et al., 2016*; *Anselmi et al., 2020*; *Blanco et al., 2008*; *Steinman, 2007*; *Steinman et al., 1975*; *Steinman*

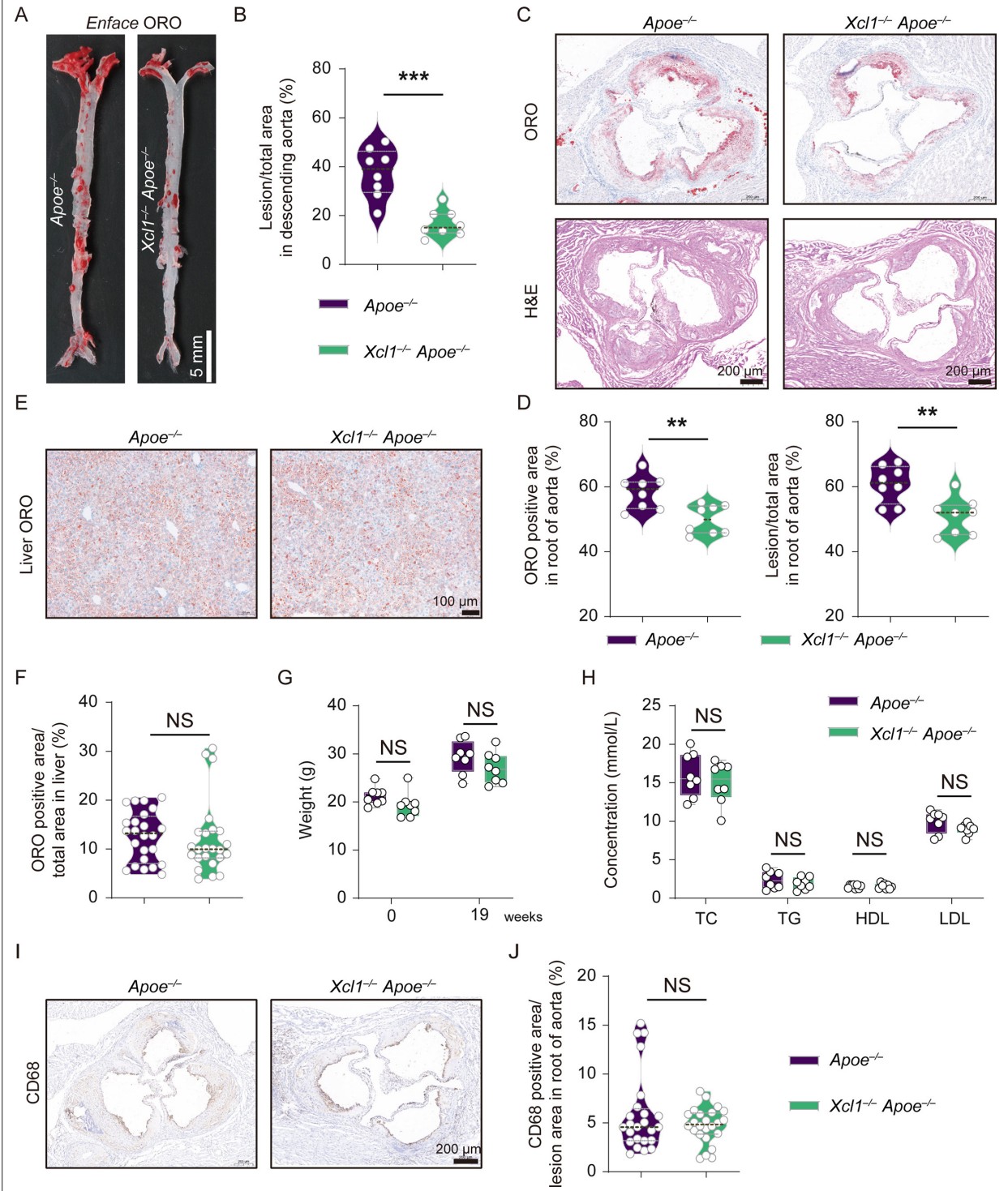

**Figure 6.** *Xcl1* deficiency reduces atherosclerotic lesion formation in *Apoe*⁻/⁻ mice. (**A**) Oil Red O (ORO) staining of the descending aortas. Scale bar, 5 mm. (**B**) Quantification of the lesion area in aortas. (**C**), ORO and H&E staining of aortic roots. Scale bar, 200 μm. (**D**) Quantification of the lesion area in aortic roots. (**E and F**) Representative liver images and quantification of ORO-positive area in livers (n=24 regions from 8 mice, 3 regions per mice). Scale bar, 100 μm. (**G**) The concentrations of total cholesterol (TC), triglycerides (TG), low-density lipoprotein (LDL), and high-density lipoprotein (HDL) in the serum. (**H**), Body weight of mice before and after feeding with 19 week high-fat diet (HFD). (**I**) Representative immunohistochemical staining of CD68 in aortic roots. (**J**) Quantification of CD68 positive area in the lesions of aortic roots (n=24 lesions from 8 mice, 3 lesions per mice). Scale bar, 100 μm. *Apoe*⁻/⁻ mice and *Xcl1*⁻/⁻ *Apoe*⁻/⁻ mice fed a HFD for 19 weeks (n=8 per group). Data represent as mean ± SEM. **p<0.01, ***p<0.001, NS, non-significant.

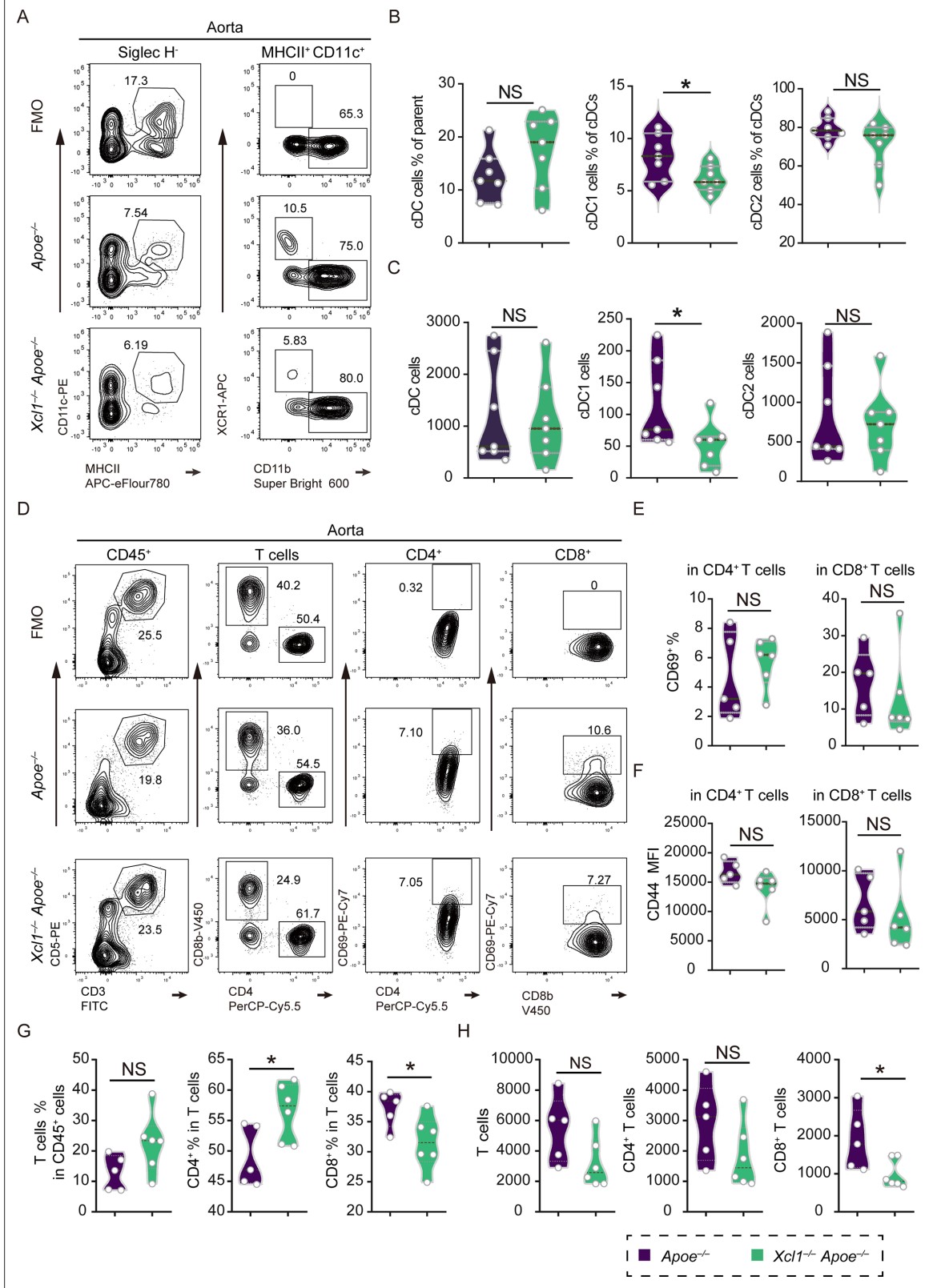

**Figure 7.** *Xcl1* deficiency reduces CD8⁺ T cells in the aortas from *Apoe*⁻/⁻ mice. (**A**) Representative flow cytometric analysis of cDC, cDC1, and cDC2 cells in aortas. (**B** and **C**) Quantification of the frequencies and absolute counts of cDC, cDC1, and cDC2 cells in aortas (n=7). (**D**) Representative flow cytometric analysis of T cell subsets in aortas, including CD3e⁺ CD5⁺, CD4⁺, CD4⁺ CD69⁺, CD8⁺, and CD8⁺ CD69⁺ T cells. (**E**) Quantification of the frequencies of CD4⁺CD69⁺ and CD8⁺ CD69⁺ in aortas. (**F**), Quantification of the Mean Fluorescence Intensity (MFI) of CD44 in CD4⁺ and CD8⁺ T cells in

*Figure 7 continued on next page*

Figure 7 continued

aortas. (**G** and **H**) Quantification of the frequencies and absolute counts of CD3e$^+$ CD5$^+$, CD4$^+$, and CD8$^+$ cells in aortas. (D-H, *Apoe*$^{-/-}$ mice (n=5) and *Xcl1*$^{-/-}$*Apoe*$^{-/-}$ mice (n=6) fed with a 16 week high-fat diet HFD). Data represent as mean ± SEM. * p<0.05, NS, non-significant.

The online version of this article includes the following figure supplement(s) for figure 7:

**Figure supplement 1.** Dendritic cell (DC) analysis in lymph nodes and spleens of *Apoe*$^{-/-}$ and *Xcl1*$^{-/-}$ *Apoe*$^{-/-}$ mice fed with 19 week high-fat diet (HFD).

**Figure supplement 2.** T cell analysis in lymph node and spleen of *Apoe*$^{-/-}$ and *Xcl1*$^{-/-}$ *Apoe*$^{-/-}$ mice fed with 16 week high-fat diet (HFD).

*et al., 1979*; *Clark et al., 2019*). Moreover, the restricted characterization of each subset of DCs in atherosclerosis, especially when compared to that in well-studied organs such as the skin and lymph nodes (*Eisenbarth, 2019*), has been severely hindered by the limited application of appropriate genetic models and in vivo data. Equally significant is that dissecting the functions of each subset of DCs in vivo relies on the development of genetic tools validated for lineage specificity. Recently, Xcr1$^+$ cDC1 has been extensively investigated using the Cre line under the promoter of the Xcr1, despite well-documented expression leakiness (*Wohn et al., 2020*; *Mattiuz et al., 2018*). Compelling evidence from tumor and viral infection models (*Bayerl et al., 2023*; *Böttcher et al., 2018*; *Brewitz et al., 2017*; *Alexandre et al., 2016*; *Domenjo-Vila et al., 2023*), as well as metabolic disorders like non-alcoholic steatohepatitis (*Deczkowska et al., 2021*), has demonstrated the critical role of Xcr1$^+$ cDC1 cells in diseases. Nevertheless, the precise role of Xcr1$^+$ cDC1 cells in atherosclerosis remains undetermined.

The precise characterization of Xcr1$^+$ cDC1 and its functions within the atherosclerotic milieu could provide valuable insights into the immunopathogenesis of the disease. By delineating the specific contributions of Xcr1$^+$ cDC1, we may identify novel therapeutic targets and develop more targeted and effective strategies for the prevention and treatment of atherosclerosis. Therefore, an investigation to understand the roles of Xcr1$^+$ cDC1 in the context of atherosclerosis and to identify therapeutic targets is highly warranted. However, at least two major obstacles impede the in vivo study of Xcr1$^+$ cDC1 in atherosclerotic mice. First, Xcr1$^+$ cDC1 cells are extremely scarce in number, and it is quite challenging to obtain highly purified Xcr1$^+$ cDC1 cells from aortic plaque in hyperlipidemic mice. Second, there are very few appropriate genetic model systems that permit functional analyses of Xcr1$^+$ cDC1 in vivo. Earlier studies on the role of Xcr1$^+$ cDC1 cells in atherosclerosis have yielded conflicting results, largely due to the use of different mouse models. For instance, the deletion of *Batf3*, *Irf8*, *Dngr1*, or *Flt3* decreased the cDC1 cells. Nevertheless, *Batf3* deficiency had no impact on atherosclerosis in hyperlipidemic mice. In contrast, *Irf8* or *Dngr1* deficiency reduced the development of atherosclerosis, while Flt3 deficiency exacerbated the condition (*Legein et al., 2015*; *Gil-Pulido et al., 2017*; *Choi et al., 2011*; *Clément et al., 2018*; *Haddad et al., 2017*). The recently developed Xcr1-Cre was found to have leakiness of recombinase activity in other immune cells (*Wohn et al., 2020*). Here, we optimized the design for the construction of the Cre line using the endogenous promoter of Xcr1 and validated its fidelity under hyperlipidemic conditions. We crossed the novel knock-in line termed *Xcr1*$^{Cre-Gfp}$ with *Rosa26*$^{LSL-tdRFP}$ Cre reporter mice and used the resulting F1 hybrids as bone marrow donors. The bone marrow cells were then transferred into *Apoe*$^{-/-}$ mice, which were subsequently fed on a HFD. We monitored the Cre recombinase activity, which was reflected by the tdRFP expression of this novel mouse line. Under hyperlipidemic conditions, we did not observe any leakiness of Cre activity, as tdRFP was strictly restricted to cDC1 cells. To the best of our knowledge, this novel Cre line is the first of its kind. We further crossed the *Xcr1*$^{Cre-Gfp}$ and *Rosa26*$^{LSL-DTA}$ mice with *Apoe*$^{-/-}$ mice, respectively. Subsequently, by further crossing, we obtained *Xcr1*$^{Cre-Gfp}$ *Rosa26*$^{LSL-DTA}$ *Apoe*$^{-/-}$ mice to create a complex genetic model, in which Xcr1$^+$ cDC1 is depleted in vivo in the context of atherosclerosis. Very interestingly, in the Xcr1$^+$ cDC1-depleted mice, atherosclerosis was significantly ameliorated. Coupled with the data we obtained from murine and human samples, where Xcr1$^+$ cDC1 cells exist in the aortic plaque and their accumulation is correlated with disease progression, we conclude that Xcr1$^+$ cDC1 is essential for the development of atherosclerosis.

Despite the fact that Xcr1$^+$ cDC1 cells are extremely rare in number, we were inclined to understand if there were any differences among Xcr1$^+$ cDC1 cells from different organs. Firstly, we determined that the Xcr1$^+$ cDC1 in the aortic plaque is replenished by circulating precursors using the bone marrow transfer model. Then we compared, under hyperlipidemic conditions, the Xcr1$^+$ cDC1 cells sorted from the spleen and lymph nodes, supplemented by a small number of but well-purified Xcr1$^+$

cDC1 cells from aortic lesions. To our surprise, the clusters of Xcr1+ cDC1 displayed obvious distinctions among different organs. When the Xcr1+ cDC1 cells from aortic lesions were projected to the clusters, we found that the major subsets present in the lymph nodes, characterized by the expression of markers *Ccl5* and *Fabp5*, and in the spleen, marked by *Stk17b* and *Gpr171*, were basically absent in the aorta. Meanwhile, compared with spleen and lymph nodes, the aortic Xcr1+ cDC1 cells are highly expressed *Ccr2*, which is an extremely important chemokine receptor dominating monocyte homing (*Mysore et al., 2022*; *Liu et al., 2022*; *Kim et al., 2023*).

This pronounced tissue-specific compartmentalization of Xcr1+ cDC1 subsets may related to multiple mechanisms, including developmental imprinting that instructs precursor differentiation into transcriptionally distinct subpopulations (*Liu et al., 2019*), and microenvironmental filtering through organ-specific chemokine axes (e.g. CCL2/CCR2 in spleen) selectively recruits receptor-matched subsets (*Bosmans et al., 2023*; *Mildner et al., 2017*). This spatial specialization optimizes pathogen surveillance for local immunological challenges. Based on the maturation analysis of the cDC1 scRNA seq data (*Bosteels et al., 2023*), our findings suggest that the aortic cDC1 cells display a major difference from those of spleen and lymph nodes by lacking the mature clusters, whereas lymph node cDC1 cells contain an additional *Fabp5+ S100 a4+* late mature Cluster. Our results also suggest that hyperlipidaemia contributes to alteration in early immature cDC1 and in the abundance of late immature cDC1 cells, which was associated with dramatic change in gene expression of *Tnfaip3*, *Serinc3*, *Apol7c*, and *Tifab*.

In other disease models, Xcr1+ cDC1 cells are crucial for activating T cells (*Alexandre et al., 2016*; *Inui et al., 2023*), and we confirmed that in hyperlipidemic mice, the depletion of cDC1 substantially attenuated T cell activation responses. It is important to note that Xcr1+ cDC1 cells are known for a well-established role in cross-presentation to CD8+ T cells. Recent studies have shown Xcr1+ cDC1 cells also engage with CD4+ T cells (*Wohn et al., 2020*; *Eickhoff et al., 2015*). Our data showing Xcr1+ cDC1-dependent CD4 and CD8 T cell activation in atherosclerotic plaques is in line with the new paradigm for cDC1s as a platform for both CD4 and CD8 T cell responses in the novel context of atherosclerosis. Considering the multifaceted roles of macrophages in lipid metabolism, inflammation, and plaque dynamics during the development of atherosclerosis (*Moore and Tabas, 2011*), we evaluated the presence of macrophages in atherosclerotic lesions using IHC and flow cytometry. Our results indicated comparable proportions of macrophages in the aorta between Xcr1+ cDC1-depleted mice and *Apoe*−/− control mice. Additionally, our results also demonstrated that the loss of Xcr1+ cDC1 cells did not affect lipid status.

Since we elucidated the critical role of Xcr1+ cDC1 cells in atherosclerosis, our aim was to test the therapeutic potential of inhibiting this specific cell type during disease progression. As Xcr1 is the sole receptor for Xcl1, and Xcl1 selectively attracts Xcr1+ cDC1 cells in both mice and humans (*Yoshida et al., 1999*; *Yoshida et al., 1998*), we performed *Xcl1* knockout on the background of *Apoe*−/− mice and assessed the impact on atherosclerosis development. We found that the Xcl1-Xcr1 axis plays a crucial role in the progression of atherosclerosis. Our results demonstrated that the loss of Xcl1 in hyperlipidemic *Apoe*−/− mice significantly reduced atherosclerotic lesion formation, decreased the frequencies and absolute counts of Xcr1+ cDC1 cells and CD8+ T cells in the aorta, without affecting lipid status or the number of aortic macrophages. Notably, while complete ablation of Xcr1+ cDC1s impaired T cell activation, reduction of Xcr1+ cDC1 recruitment via Xcl1 deletion did not significantly compromise this process. This discrepancy may arise through compensatory mechanisms: alternative chemokine axes (e.g. CCL5/CCR5, CXCL9/CXCR3, BCL9/BCL9L) may partially rescue Xcr1+ cDC1 homing (*Eisenbarth, 2019*; *de Oliveira et al., 2017*; *He et al., 2024*), while non-cDC1 antigen-presenting cells (e.g. monocytes, cDC2s) may sustain T cell activation (*Brewitz et al., 2017*; *Böttcher et al., 2015*). Furthermore, tissue-specific microenvironment factors could potentially modulate its role in other diseases. In summary, our findings identify Xcl1 as a potential therapeutic target for atherosclerosis therapy, though its cellular origins and regulation of lesional Xcr1+ cDC1 and T cells dynamics require further studies.

## Materials and methods

### Animal experiments and patients

All mice were raised in a Specific Pathogen-Free facility and the experiments were conducted in accordance with the guidelines approved by the Animal Ethics Committee of Xinxiang Medical University (China) (Reference No. XYLL-2016S001). The sections of the human femoral artery were obtained in accordance with our previous study (*Li et al., 2020*) and approved by the Ethics Committee of the Third Military Medical University (Project identification code SYXK-PLA-20170005). *Rosa26$^{LSL-RFP}$* knock-in mice were generously provided by Hervé Luche at University Clinics Ulm in Germany (*Luche et al., 2007*). C57BL/6 and *Apoe$^{-/-}$* mice were purchased from Beijing Vital River Laboratory Animal Technology (China). *Rosa26$^{LSL-DTA}$* knock-in mice (Stock Number 009669) were purchased from the Jackson Laboratory (United States of America). Both HFD (contained 20% protein, 20% carbohydrate, 40% fat, and 1.25% cholesterol, Cat: XT108C), and ND (contained 20% protein, 64% carbohydrate, 20% fat, Cat: AIN-93G) were purchased from Jiangsu Xietong Pharmaceutical Bioengineering Co., Ltd. 7–8 weeks old mice were used to establish an atherosclerosis model through fed on a HFD for 8–20 weeks. On harvest, each sample was assigned an analytical code that was irrelevant to its genotype and processed and analyzed by a researcher blinded to its origin.

### Generation of *Xcr1$^{Cre-Gfp}$* knock-in mice

The *Xcr1$^{Cre-Gfp}$* mouse line was generated by inserting a Cre-P2A-GFP-P2A cassette immediately upstream of the endogenous *Xcr1* translational start codon (ATG). This design enables co-expression of Cre recombinase and GFP under the control of the native *Xcr1* promoter. A 3.2 kb repair template was synthesized (GenScript, Nanjing, China), consisting of a 0.62 kb 5' homology arm (HA), a codon-optimized *Cre* sequence fused to a P2A peptide, an enhanced GFP reporter linked to a second P2A site, and a 0.67 kb 3' homology arm. For targeted integration, CRISPR/Cas9-mediated homology-directed repair (HDR) was performed by co-injecting Cas9 mRNA, a chimeric sgRNA (sgXcr1-KI), and the repair template into fertilized C57BL/6 zygotes at the one-cell stage, following established protocols (*Chu et al., 2016*; *Larson, 2020*). Founders (F0) harboring the correct knock-in allele were identified by PCR and Sanger sequencing, then used to establish heterozygous lines. The sgRNA sequence is provided in *Supplementary file 3*.

### Breeding strategy of *Xcr1$^{Cre-Gfp}$* *Rosa26$^{LSL-DTA}$* *Apoe$^{-/-}$*

To generate *Xcr1$^{Cre-Gfp}$* *Rosa26$^{LSL-DTA}$* *Apoe$^{-/-}$* mice, we performed the following crosses: *Xcr1$^{Cre-Gfp}$* mice were crossed with *Apoe$^{-/-}$* mice, and offspring heterozygous for both alleles were intercrossed to obtain homozygous *Xcr1$^{Cre-Gfp}$* *Apoe$^{-/-}$* mice. *Rosa26$^{LSL-DTA}$* mice were crossed with *Apoe$^{-/-}$* mice, and progeny heterozygous for both alleles were intercrossed to generate homozygous *Rosa26$^{LSL-DTA}$* *Apoe$^{-/-}$* mice. Finally, homozygous *Xcr1$^{Cre-Gfp}$* *Apoe$^{-/-}$* mice were crossed with homozygous *Rosa26$^{LSL-DTA}$* *Apoe$^{-/-}$* mice to produce *Xcr1$^{Cre-Gfp}$* *Rosa26$^{LSL-DTA}$* *Apoe$^{-/-}$* offspring.

### Generation of *Xcl1* knockout mice

The *Xcl1* knockout mouse line was generated using CRISPR/Cas9 genome editing through the microinjection of a mixture containing Cas9 mRNA and two sgRNAs into fertilized C57BL/6 zygotes at the one-cell stage, as previously described (*Voisinne et al., 2019*). The genotypes were confirmed using PCR and Sanger sequencing. Subsequently, *Xcl1$^{-/-}$* and *Apoe* double knockout mice were obtained by crossing the *Xcl1$^{-/-}$* mice with *Apoe$^{-/-}$* mice. The sequences of the sgRNA are reported in *Supplementary file 3*.

### Bone marrow transplantation

After one week of antibiotic treatment, the recipient CD45.1$^+$ CD45.2$^+$*Apoe$^{-/-}$* mice were injected intraperitoneally with busulfan at a dosage of 30 µg/g every two days for one week. Subsequently, bone marrow from 8-week-old CD45.2$^+$*Xcr1$^{Cre-Gfp}$* *Rosa26$^{LSL-RFP}$*, CD45.2$^+$*Xcr1$^{Cre-Gfp}$* *Rosa26$^{LSL-DTA}$*, or WT (CD45.2$^+$ wild-type) donor mice was intravenously transferred to each recipient mouse, with two million cells administered per recipient. The efficiency of transplantation was assessed 4-week post-transplantation, after which all mice were fed a HFD for 16 weeks. Finally, the mice were sacrificed

for *en face* analysis of the descending aorta, hematoxylin and eosin (H&E) staining, Oil Red O (ORO) staining of the aortic root, and flow cytometry analysis, as previously described (*Huang et al., 2019*).

## Immunofluorescence

Frozen sections of the aortic root from mice were incubated at 60 °C for 2 min and then fixed in 4% paraformaldehyde for 30 min. The samples were subsequently washed with PBS and permeabilized/blocked with 0.1% Triton X-100 in 5% BSA. Next, the primary antibodies, including anti-Xcr1 (BD Biosciences, Cat. 148212) and anti-CD11c (BD Biosciences, Cat. 12-0114-83), were applied in a wet box at 4 °C overnight. After washing with PBST (PBS containing 0.1% Tween-20) for five times, the samples were incubated with secondary antibodies for 1 hr, followed by a 10 min incubation with DAPI at room temperature (protected from light). The fluorescent signals of the sections were detected using a Nikon C2 confocal microscope (Nikon, Japan).

## Morphometric and immunohistochemical analyses

After modeling, the heart and aorta tissue were obtained from mice. The *en face* descending aortas were fixed with 4% paraformaldehyde and subsequently stained with ORO. Additionally, frozen sections of the aortic roots were stained with ORO and H&E to assess the severity of atherosclerosis, and frozen sections of livers were stained with ORO to assess the severity of lipid accumulation, in accordance with our previous study (*Huang et al., 2019*). Paraffin-embedded sections of the aortic root were dewaxed in xylene twice for 10 min, rehydrated through descending grades of ethanol, and subjected to sodium citrate-induced antigen retrieval for 10 min at 95 °C. The sections were then incubated with 3% $H_2O_2$ for 10 min at room temperature. Following this, the sections were blocked and incubated with primary antibodies, either anti-Xcr1 (NBP1-02343) or CD68 (FB113109). After five washes with DPBS, the sections were incubated with secondary antibodies for 30 min and stained with DAB at room temperature. Subsequently, the sections were counterstained with hematoxylin, dehydrated in ethanol, and mounted with neutral balsam. Images were captured using the digital pathology system (Pannoramic MIDI, Hungary), viewed using CaseViewer2.4 software and analyzed using ImageJ software.

## Flow cytometry analysis

To obtain the single-cell suspension, aortas, lymph nodes and spleens were collected from each mouse. Spleens were directly dissociated using the Gentle MACS Dissociators (Miltenyi Biotec). Aortas and lymph nodes were cut into small pieces and then dissociated using the Gentle MACS Dissociators. Dissociated tissues digested with digestive enzymes (Collagenase IV, Cat: V900893 from Sigma; DNase I, Cat: DN25 from Sigma; Liberase DH, Cat: 5401054001 from Roche) for 30 min at 37 °C using a shaker set to 300 rpm or a heating block. The resulting cell suspension was filtered through 75 μm strainers and subsequently blocked with 2.4G2 for 20 min at 4 °C, followed by washing with stain buffer (BD, Cat: 554656). Then samples were stained with monoclonal antibody mixes for 30 min at 4 °C followed by washing with stain buffer. Finally, cells were resuspended in stain buffer containing 200 nM Sytox blue (Thermo Fisher Scientific, Cat: S34857) and acquired using a flow cytometer (Thermo Fisher Scientific, Invitrogen Attune NxT Flow Cytometer), and the FACS data were analyzed using FlowJo software. The following antibodies purchased from BioLegend, BD or eBioscience were used to detect surface markers by flow cytometry: anti-CD45.2 (104) conjugated to eFluor 450, FITC or PE-Cy7; anti-CD45.1 (A20) conjugated to PE or eFluor450; anti-CD45 (30-F11) conjugated to PE; anti-CD11b (M1/70) conjugated to Super Bright 600, BUV395 or biotin; anti-CD3e (145–2 C11) conjugated to PE-Cy5.5 or PE-Cy7; anti-CD3e (eBio500A2) conjugated to Alexa Fluor700; anti-CD19 (eBio1D3) conjugated to PE-Cy5.5 or biotin; anti-Xcr1 (ZET) conjugated to APC or BV650; anti-Ly6G (1A8) conjugated to Alexa Fluor 700; anti-MHCII (M5/114.15.2) conjugated to APC-eFluor780; anti-CD11c (N418) conjugated to PE or PE-Cy5.5; anti-Siglec H (eBio440c) conjugated to PE-Cy7 or PE; anti-CD8b (eBioH35-17.2) conjugated to eFluor 450; anti-CD44 (IM7) conjugated to BV605; anti-CD4 (RM4-5) conjugated to PE-Cy5.5; anti-CD69 (H1.2F3) conjugated to PE-Cy7; anti-CD62L (MEL-14) conjugated to APC; anti-CD5 (53–7.3) conjugated to APC-eFluor780 or biotin; anti-Siglec F (E50-2440) conjugated to BV421; anti-F4/80 (BM8) conjugated to PE-Cy7; anti-CD64 (X54-5/7.1) conjugated to APC; anti-NK1.1 (PK136) conjugated to Alexa Fluor 700; anti-CD172α (P84) conjugated to FITC.

## Single-cell RNA sequencing

Eight-week-old male *Apoe*$^{-/-}$ mice were fed with 20 week ND or HFD. The single cell suspension of spleen, lymph node, and aorta were obtained according to the flow cytometry analysis section. Single-cell suspension from spleens and lymph nodes were firstly removed the CD5$^+$, CD19$^+$, and CD11b$^+$ cells via SAV conjugated beads. Then two million live cells from spleen and lymph node were stained with mix1 (CD45.2-PE-Cy7, CD11c-PE-Cy5.5, CD172α-FITC, MHCII- APC-eFluor780, and Xcr1-APC), and all cells from aorta were stained with mix2 (CD45.2-PE-Cy7, CD11b-super bright 600, CD11c-PE-Cy5.5, MHCII- FITC, CD3- Alexa Fluor 700, CD19- Alexa Fluor 700, Ly6G- Alexa Fluor 700, and Xcr1-APC) for 30 min at 4 °C. After washing with 1 mL FACS buffer two times, cells from different organs were stained with different anti-CD45 antibodies conjugated with barcode oligos from BD Ms Single Cell Sample Multiplexing Kit (BD Bioscience, Cat: 633793). Finally, cells were resuspended with Sytox blue and CD11c$^+$ MHCII$^+$ Xcr1$^+$ cells were sorted and subjected to the BD Rhapsody Express system. Then, cDNA and sample tag libraries were built with BD Rhapsody whole transcriptome amplification (WTA) reagent kit (BD Bioscience, Cat: 633733, 633773, 633801, 664887) following manufacturer's instructions, and sequenced on an illumina Novaseq 600 sequencer. Pair-end Fastq files of sample tag and WTA data were processed via BD Rhapsody analysis pipeline v2.0, and the resultant dataset was mainly analyzed using SeqGeq software, which contains Lex-BDSMK and Seurat v4.04 plugin components. This experiment was performed once, and both raw data and processed data were uploaded into GEO (Accession number: GSE279370).

## Statistical analysis

All collected data were included and presented as means ± SEM (Standard Error of the Mean), reflecting the average values of biological measurements across multiple samples. Data analysis was performed using GraphPad Prism software (version 8.0). Statistical significance was assessed by both parametric (unpaired Student's *t*-test) and nonparametric (Mann–Whitney test) methods when comparing two groups. The assumptions of the tests were checked to ensure appropriate application. Significant differences between different groups were set at *$p < 0.05$, **$p < 0.01$, ***$p < 0.001$, and ****$p < 0.0001$. Graphical representations of the data will include bar graphs with error bars denoting SEM.

## Acknowledgements

We thank the animal facility and flow cytometry facility of GRIS in Xinxiang Medical University for providing technical support and assistance. Generation of murine genetic models were supported by 111 program (D20036). This research was funded by grants from the Science and Technology Innovation Program of Hunan Province [2023DK2005], [2024RC1071], and [2022RC1223], and NSFC grants 32170879 and 82301972, and the Science and Technology Department of Henan Province 242102310030.

## Additional information

### Funding

| Funder | Grant reference number | Author |
| --- | --- | --- |
| Science and Technology Innovation Program of Hunan Province | 2023DK2005 | Yinming Liang |
| Science and Technology Innovation Program of Hunan Province | 2024RC1071 | Yinming Liang |
| Science and Technology Innovation Program of Hunan Province | 2022RC1223 | Lichen Zhang |
| National Natural Science Foundation of China | 32170879 | Lichen Zhang |

| Funder | Grant reference number | Author |
| --- | --- | --- |
| National Natural Science Foundation of China | 82301972 | Tianhan Li |
| Science and Technology Department of Henan Province | 242102310030 | Tianhan Li |

The funders had no role in study design, data collection and interpretation, or the decision to submit the work for publication.

## Author contributions

Tianhan Li, Data curation, Software, Funding acquisition, Visualization, Writing – original draft; Liaoxun Lu, Data curation, Writing – original draft; Juanjuan Qiu, Yanrong Gu, Data curation; Xin Dong, Tingting Jia, Data curation, Software; Le Yang, Validation; Kexin He, Software; Binhui Zhou, Software, Supervision; Toby Lawrence, Hui Wang, Methodology, Writing – review and editing; Marie Malissen, Writing – review and editing; Guixue Wang, Resources, Writing – review and editing; Rong Huang, Data curation, Supervision, Writing – review and editing; Bernard Malissen, Resources, Supervision, Methodology, Project administration, Writing – review and editing; Yinming Liang, Conceptualization, Resources, Funding acquisition, Methodology, Writing – original draft, Project administration; Lichen Zhang, Resources, Funding acquisition, Writing – original draft, Writing – review and editing

## Author ORCIDs

Bernard Malissen [iD] https://orcid.org/0000-0003-1340-9342
Yinming Liang [iD] https://orcid.org/0000-0001-9174-4037
Lichen Zhang [iD] https://orcid.org/0000-0002-7810-9941

## Ethics

Human subjects: The sections of the human femoral artery were obtained in accordance with our previous study and approved by the Ethics Committee of the Third Military Medical University (Project identification code SYXK-PLA-20170005).
All mice were raised in a Specific Pathogen-Free facility and the experiments were conducted in accordance with the guidelines approved by the Animal Ethics Committee of Xinxiang Medical University (China) (Reference No. XYLL-2016S001).

Reviewer #1 (Public review): https://doi.org/10.7554/eLife.107742.3.sa1
Reviewer #2 (Public review) https://doi.org/10.7554/eLife.107742.3.sa2
Author response https://doi.org/10.7554/eLife.107742.3.sa3

# Additional files

## Supplementary files

Supplementary file 1. The top 10 highly expressed marker genes in ten clusters.

Supplementary file 2. Upregulated genes in Xcr1+ cDC1 cells from aorta vs spleen, and aorta vs lymph node in *Apoe*−/− mice fed a high-fat diet (HFD) for 20 weeks.

Supplementary file 3. The sequences of the sgRNA used for generation of *Xcr1*Cre-Gfp knock-in mice and *Xcl1* knockout mice.

MDAR checklist

## Data availability

Raw data and processed data have been deposited in GEO under accession codes GSE279370.

The following dataset was generated:

| Author(s) | Year | Dataset title | Dataset URL | Database and Identifier |
|---|---|---|---|---|
| Liang Y | 2025 | Xcr1+ type 1 conventional dendritic cells are essential mediators for atherosclerosis progression | http://www.ncbi.nlm.nih.gov/geo/query/acc.cgi?acc=GSE279370 | NCBI Gene Expression Omnibus, GSE279370 |

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
