## [Editor Report · eLife Assessment]

This manuscript by Li, Lu et al., presents **important** findings on the role of cDC1 in atherosclerosis and their influence on the adaptive immune system. Using Xcr1Cre-Gfp Rosa26LSL-DTA ApoE-/- mouse models, these data **convincingly** reveal an unexpected, non-redundant role of the XCL1-XCR1 axis in mediating cDC1 contributions to atherosclerosis.

---

## [Referee Report · Reviewer #1 (Public review)]

Summary:

In this study by Li et al., the authors re-investigated the role of cDC1 for atherosclerosis progression using the ApoE model. First, the authors confirmed the accumulation of cDC1 in atherosclerotic lesions in mice and humans. Then in order to examine the functional relevance of this cell type, the authors developed a new mouse model to selectively target cDC1. Specifically, they inserted the Cre recombinase directly after the start codon of endogenous XCR1 gene, thereby avoiding off-target activity. Following validation of this model, the authors crossed it with ApoE-deficient mice and found a striking reduction of aortic lesions (numbers and size) following high fat diet. The authors further characterized the impact of cDC1 depletion on lesional T cells and their activation state. Also, they provide in-depth transcriptomic analyses of lesional in comparison to splenic and nodal cDC1. These results imply cellular interactions between lesion T cells and cDC1. Finally, the authors show that the chemokine XCL1, which is produced by activated CD8 T cells (and NK cells) plays a key role for the interaction with XCR1-expressing cDC1 and particularly for the atherosclerotic disease progression.

Strengths:

The surprising results on XCL1 represent a very important gain in knowledge. The role of cDC1 is clarified with a new genetic mouse model.

Comments on revised version:

The authors have addressed my concerns in the revised version of this manuscript.

---

## [Referee Report · Reviewer #2 (Public review)]

This study investigates the role of cDC1 in atherosclerosis progression using Xcr1Cre-Gfp Rosa26LSL-DTA ApoE-/- mice. The authors demonstrate that selective depletion of cDC1 reduces atherosclerotic lesions in hyperlipidemic mice. While cDC1 depletion did not alter macrophage populations, it suppressed T cell activation (both CD4+ and CD8+ subsets) within aortic plaques. Further, targeting the chemokine Xcl1 (ligand of Xcr1) effectively inhibits atherosclerosis. The manuscript is well-written, and data are clearly presented. The data provided in the article can well support the author's conclusion.

Comments on revised version:

The authors have addressed all previous concerns and made appropriate revisions to the data. I have no further questions.

---

## [Author Response]

The following is the authors’ response to the original reviews.

**Reviewer #1 (Public review):**
Summary:In this study by Li et al., the authors re-investigated the role of cDC1 for atherosclerosis progression using the ApoE model. First, the authors confirmed the accumulation of cDC1 in atherosclerotic lesions in mice and humans. Then, in order to examine the functional relevance of this cell type, the authors developed a new mouse model to selectively target cDC1. Specifically, they inserted the Cre recombinase directly after the start codon of the endogenous XCR1 gene, thereby avoiding off-target activity. Following validation of this model, the authors crossed it with ApoE-deficient mice and found a striking reduction of aortic lesions (numbers and size) following a high-fat diet. The authors further characterized the impact of cDC1 depletion on lesional T cells and their activation state. Also, they provide in-depth transcriptomic analyses of lesional in comparison to splenic and nodal cDC1. These results imply cellular interactions between lesion T cells and cDC1. Finally, the authors show that the chemokine XCL1, which is produced by activated CD8 T cells (and NK cells), plays a key role in the interaction with XCR1-expressing cDC1 and particularly in the atherosclerotic disease progression.Strengths:The surprising results on XCL1 represent a very important gain in knowledge. The role of cDC1 is clarified with a new genetic mouse model.

Thank you

Weaknesses:My criticism is limited to the analysis of the scRNAseq data of the cDC1. I think it would be important to match these data with published data sets on cDC1. In particular, the data set by Sophie Janssen's group on splenic cDC1 might be helpful here (PMID: 37172103; https://www.single-cell.be/spleen_cDC_homeostatic_maturation/datasets/cdc1). It would be good to assign a cluster based on the categories used there (early/late, immature/mature, at least for splenic DC).

Thank you very much for your help. Using the scRNA seq data of Xcr1^+^ cDC1 sorted from ApoE^–/–^ mice, we re-annotated the populations, following the methodology proposed by Sophie Janssen's group. These results are presented in Figure S9 and Figure S10 and described in detail in the Results and Discussion section.

Please refer to the Results section from line 264 to 284: “Using the scRNA seq data of Xcr1^+^ cDC1 sorted from hyperlipidemic mice, we annotated the 10 populations as shown in Figure S9A, following the methodology from a previous study [41]. Ccr7^+^ mature cDC1s (Cluster 3, 7 and 9) and Ccr7- immature cDC1s (remaining clusters) were identified across cDC1 cells sorted from aorta, spleen and lymph nodes (Figure S9B). Further stratification based on marker genes reveals that Cluster 10 is the pre-cDC1, with high expression level of CD62L (Sell) and low expression level of CD8a (Figure S9C). Cluster 6 and 8 are the proliferating cDC1s, which express high level of cell cycling genes Stmn1 and Top2a (Figure S9D). Cluster 1 and 4 are early immature cDC1s, and cluster 2 and 5 are late immature cDC1s, according to the expression pattern of Itgae, Nr4a2 (Figure S9E). Cluster 9 cells are early mature cDC1s, with elevated expression of Cxcl9 and Cxcl10 (Figure S9F). Cluster 3 and 7 as late mature cDC1s, characterized by the expression of Cd63 and Fscn1 (Figure S9G). As shown in Figure 5C and Figure S9, the 10 populations displayed a major difference of aortic cDC1 cells that lack in pre-cDC1s (cluster 10) and mature cells (cluster 3, 7 and 9). Interestingly, in hyperlipidemic mice splenic cDC1 possess only Cluster 3 as the late mature cells while the lymph node cDC1 cells have two late mature populations namely Cluster 3 and Cluster 7. In further analysis, we also compared splenic cDC1 cells from HFD mice to those from ND mice. As shown in Figure S10, HFD appears to impact early immature cDC1-1 cells (Cluster 1) and increases the abundance of late immature cDC1 cells (Cluster 2 and 5), regardless of the fact that all 10 populations are present in two origins of samples. We also found that Tnfaip3 and Serinc3 are among the most upregulated genes, while Apol7c and Tifab are downregulated in splenic cDC1 cells sorted from HFD mice”.

Please refer to the Discussion section from line 380 to 385: “Based on the maturation analysis of the cDC1 scRNA seq data [41], our findings suggest that the aortic cDC1 cells display a major difference from those of spleen and lymph nodes by lacking the mature clusters, whereas lymph node cDC1 cells contain an additional Fabp5^+^ S100a4^+^ late mature Cluster. Our results also suggest that hyperlipidemia contributes to alteration in early immature cDC1 and in the abundance of late immature cDC1 cells, which was associated with dramatic change in gene expression of Tnfaip3, Serinc3, Apol7c and Tifab”.

**Reviewer #2 (Public review):**
This study investigates the role of cDC1 in atherosclerosis progression using Xcr1Cre-Gfp Rosa26LSL-DTA ApoE-/- mice. The authors demonstrate that selective depletion of cDC1 reduces atherosclerotic lesions in hyperlipidemic mice. While cDC1 depletion did not alter macrophage populations, it suppressed T cell activation (both CD4+ and CD8+ subsets) within aortic plaques. Further, targeting the chemokine Xcl1 (ligand of Xcr1) effectively inhibits atherosclerosis. The manuscript is well-written, and the data are clearly presented. However, several points require clarification:(1) In Figure 1C (upper plot), it is not clear what the Xcr1 single-positive region in the aortic root represents, or whether this is caused by unspecific staining. So I wonder whether Xcr1 single-positive staining can reliably represent cDC1. For accurate cDC1 gating in Figure 1E, Xcr1+CD11c+ co-staining should be used instead.

The observed false-positive signal in the wavy structures within immunofluorescence Figure 1C (upper panel) results from the strong autofluorescence of elastic fibers, a major vascular wall component (alongside collagen). This intrinsic property of elastic fibers is a well-documented confounder in immunofluorescence studies [A, B].

In contrast, immunohistochemistry (IHC) employs an enzymatic chromogenic reaction (HRP with DAB substrate) that generates a brown precipitate exclusively at antigen-antibody binding sites. Importantly, vascular elastic fibers lack endogenous enzymatic activity capable of catalyzing the DAB reaction, thereby preventing this source of false positivity in IHC.

Given that Xcr1 is exclusively expressed on conventional type 1 dendritic cells [C], and considering that IHC lacks the multiplexing capability inherent to immunofluorescence for antigen co-localization, single-positive Xcr1 staining reliably identifies cDC1s in IHC results.

[A] König, K et al. “Multiphoton autofluorescence imaging of intratissue elastic fibers.” Biomaterials vol. 26,5 (2005): 495-500. doi:10.1016/j.biomaterials.2004.02.059

[B] Andreasson, Anne-Christine et al. “Confocal scanning laser microscopy measurements of atherosclerotic lesions in mice aorta. A fast evaluation method for volume determinations.” Atherosclerosis vol. 179,1 (2005): 35-42. doi:10.1016/j.atherosclerosis.2004.10.040

[C] Dorner, Brigitte G et al. “Selective expression of the chemokine receptor XCR1 on cross-presenting dendritic cells determines cooperation with CD8+ T cells.” Immunity vol. 31,5 (2009): 823-33. doi:10.1016/j.immuni.2009.08.027

(2) Figure 4D suggests that cDC1 depletion does not affect CD4+/CD8+ T cells. However, only the proportion of these subsets within total T cells is shown. To fully interpret effects, the authors should provide:(a) Absolute numbers of total T cells in aortas.(b) Absolute counts of CD4+ and CD8+ T cells.

Thanks for your suggestions. We agree that assessing both proportions and absolute numbers in Figure 4 provides a more complete picture of the effects of cDC1 depletion on T cell populations. Furthermore, we also add the absolute count of cDC1 cells and total T cells, and CD44 MFI (mean fluorescence intensity) in CD4^+^ and CD8^+^ T cells in Figure 4, and supplemented corresponding textual descriptions in the revised manuscript.

Please refer to the Results section from line 183 to 187: “Subsequently, we assessed T cell phenotype in the two groups of mice. While neither the frequencies nor absolute counts of aortic CD4^+^ and CD8^+^ T cells differed significantly between two groups of mice (Figure 4D-F), CD69 frequency and CD44 MFI (Mean Fluorescence Intensity), the T cell activation markers, were significantly reduced in both CD4^+^ and CD8^+^ T cells from Xcr1^+^ cDC1 depleted mice compared to controls (Figure 4G and H)”.

(3) How does T cell activation mechanistically influence atherosclerosis progression? Why was CD69 selected as the sole activation marker? Were other markers (e.g., KLRG1, ICOS, CD44) examined to confirm activation status?

We sincerely appreciate these insightful comments. As extensively documented in the literature, activated effector T cells (both CD4+ and CD8+) critically promote plaque inflammation and instability through their production of pro-inflammatory cytokines (particularly IFN-γ and TNF-α), which drive endothelial activation, exacerbate macrophage inflammatory responses, and impair smooth muscle cell function [A].

In our study, we specifically investigated the role of cDC1 cells in atherosclerosis progression. Our key findings demonstrate that cDC1 depletion attenuates T cell activation (as shown by reduced CD69/CD44 expression) and that this reduction in activation is functionally linked to the observed decrease in atherosclerosis burden in our model.

Regarding CD44 as an activation marker, we performed quantitative analyses of CD44 mean fluorescence intensity (MFI) in aortic T cells (Figure 4). Importantly, the MFI of CD44 was significantly lower on both CD4+ and CD8+ T cells from Xcr1^Cre-Gfp^ Rosa26^LSL-DTA^ ApoE^–/–^ mice compared to the control ApoE^–/–^ mice (data shown below), which is consistent with the result of CD69 in Figure 4. We added the related description in the Result section.

Please refer to the Results section from line 185 to 187 “CD69 frequency and CD44 MFI (Mean Fluorescence Intensity), the T cell activation markers, were significantly reduced in both CD4+ and CD8+ T cells from Xcr1+ cDC1 depleted mice compared to controls (Figure 4G and H)”.

Similarly, MFI of CD44 was significantly lower on both CD4^+^ and CD8^+^ T cells from Xcl1^–/–^ ApoE^–/–^ mice compared to the control ApoE^–/–^ mice (data shown below), which is consistent with the result of CD69 in Figure 7. We also added the related description in the Result section.

Please refer to the Results section from line 308 to 309 “Crucially, CD69^+^ frequency and CD44 MFI remained comparable in both aortic CD4^+^ and CD8^+^ T cells between two groups (Figure 7D-F).”

[A] Hansson, Göran K, and Andreas Hermansson. “The immune system in atherosclerosis.” Nature immunology vol. 12,3 (2011): 204-12. doi:10.1038/ni.2001

(4) Figure 7B: Beyond cDC1/2 proportions within cDCs, please report absolute counts of: Total cDCs, cDC1, and cDC2 subsets. Figure 7D: In addition to CD4+/CD8+ T cell proportions, the following should be included:(a) Total T cell numbers in aortas(b) Absolute counts of CD4+ and CD8+ T cells.

Thanks for your suggestions. We have now included in Figure 7 the absolute counts of cDC, cDC1, and cDC2 cells, along with CD4^+^ and CD8^+^ T cells in aortic tissues. Additionally, we provide the corresponding CD44 mean fluorescence intensity (MFI) measurements for both CD4^+^ and CD8^+^ T cell populations. We added the related description in the Result section.

Please refer to the Results section from line 303 to 311: “The flow cytometric results illustrated that both frequencies and absolute counts of Xcr1^+^ cDC1 cells in the aorta were significantly reduced, but cDCs and cDC2 cells from Xcl1^–/–^ ApoE^–/–^ were comparable with that from ApoE^–/–^ (Figure 7A-C). Moreover, in both lymph node and spleen, the absolute numbers of pDC, cDC1 and cDC2 from Xcl1^–/–^ ApoE^–/–^ were comparable with that from ApoE^–/–^ (Figure S11). Crucially, CD69^+^ frequency and CD44 MFI remained comparable in both aortic CD4^+^ and CD8^+^ T cells between two groups (Figure 7D-F). However, aortic CD8^+^ T cells exhibited reduced frequency and absolute count, while CD4^+^ T cells showed increased frequency but unchanged counts in Xcl1^–/–^ ApoE^–/–^ mouse versus controls (Figure 7G and H).”

(5) cDC1 depletion reduced CD69+CD4+ and CD69+CD8+ T cells, whereas Xcl1 depletion decreased Xcr1+ cDC1 cells without altering activated T cells. How do the authors explain these different results? This discrepancy needs explanation.

We sincerely appreciate your professional and insightful comments regarding the mechanistic relationship between cDC1 depletion and T cell activation. Direct cDC1 depletion in the Xcr1^Cre-Gfp^ Rosa26^LSL-DTA^ ApoE^–/–^ micmodel removes both recruited and tissue-resident cDC1s, eliminating their multifunctional roles in antigen presentation, co-stimulation and cytokine secretion essential for T cell activation. In contrast, Xcl1 depletion reduces, but does not eliminate cDC1 migration into plaques. Furthermore, alternative chemokine axes (e.g., CCL5/CCR5, CXCL9/CXCR3, BCL9/BCL9L) may partially rescue cDC1 recruitment [13, 68, 69], and non-cDC1 APCs (e.g., monocytes, cDC2s) may compensate for T cell activation [55, 70]. We emphasize that Xcl1 depletion specifically failed to alter T cell activation in hyperlipidemic ApoE^–/–^ mice. However, its impact may differ in other pathophysiological contexts due to compensatory mechanisms. We thank you again for highlighting this nuance, which strengthens our mechanistic interpretation. We have added these points to the discussion section and included new references.

Please refer to the Discussion section from line 407 to 413: “Notably, while complete ablation of Xcr1^+^ cDC1s impaired T cell activation, reduction of Xcr1^+^ cDC1 recruitment via Xcl1 deletion did not significantly compromise this process. This discrepancy may arise through compensatory mechanisms: alternative chemokine axes (e.g., CCL5/CCR5, CXCL9/CXCR3, BCL9/BCL9L) may partially rescue Xcr1^+^ cDC1 homing [13, 68, 69], while non-cDC1 antigen-presenting cells (e.g., monocytes, cDC2s) may sustain T cell activation [55, 70]. Furthermore, tissue-specific microenvironment factors could potentially modulate its role in other diseases.”. [13] Eisenbarth, S C. “Dendritic cell subsets in T cell programming: location dictates function.” Nature reviews. Immunology vol. 19,2 (2019): 89-103. doi:10.1038/s41577-018-0088-1 [55] Brewitz, Anna et al. “CD8+ T Cells Orchestrate pDC-XCR1+ Dendritic Cell Spatial and Functional Cooperativity to Optimize Priming.” Immunity vol. 46,2 (2017): 205-219. doi:10.1016/j.immuni.2017.01.003 [68] de Oliveira, Carine Ervolino et al. “CCR5-Dependent Homing of T Regulatory Cells to the Tumor Microenvironment Contributes to Skin Squamous Cell Carcinoma Development.” Molecular cancer therapeutics vol. 16,12 (2017): 2871-2880. doi:10.1158/1535-7163.MCT-17-0341.[69] He F, Wu Z, Liu C, Zhu Y, Zhou Y, Tian E, et al. Targeting BCL9/BCL9L enhances antigen presentation by promoting conventional type 1 dendritic cell (cDC1) activation and tumor infiltration. Signal Transduct Target Ther. 2024;9(1):139. Epub 2024/05/30. doi: 10.1038/s41392-024-01838-9. PubMed PMID: 38811552; PubMed Central PMCID: PMCPMC11137111.[70] Böttcher, Jan P et al. “Functional classification of memory CD8(+) T cells by CX3CR1 expression.” Nature communications vol. 6 8306. 25 Sep. 2015, doi:10.1038/ncomms9306.

**Reviewer #1 (Recommendations for the authors):**
(1) Line 32 - The authors might want to add that the mouse model leads to a "constitutive" depletion of cDC1.

Thanks for your advice, we have revised the sentence as follows.

Please refer to the Results section from line 31 to 33: “we established Xcr1^Cre-Gfp^ Rosa26^LSL-DTA^ ApoE^–/–^ mice, a novel and complex genetic model, in which cDC1 was constitutively depleted in vivo during atherosclerosis development”.

(2) Line 187-188: The authors claim that T cell activation was "inhibited" if cDC1 was depleted. The data shows that the T cells were less activated, but there is no indication of any kind of inhibition; this should be corrected.

Thanks for your advice, we have revised the sentence as follows.

Please refer to the Results section from line 183 to 187: “Subsequently, we assessed T cell phenotype in the two groups of mice. While neither the frequencies nor absolute counts of aortic CD4^+^ and CD8^+^ T cells differed significantly between two groups of mice (Figure 4D-F), CD69 frequency and CD44 MFI (Mean Fluorescence Intensity), the T cell activation markers, were significantly reduced in both CD4^+^ and CD8^+^ T cells from Xcr1^+^ cDC1 depleted mice compared to controls (Figure 4G and H)”.

(3) Why are some splenic DC clusters absent in LNs and vice versa? This is not obvious to this reviewer and should at least be discussed.

We appreciate the insightful question regarding the absence of certain splenic DC clusters in LNs. This phenomenon in Figure 5 aligns with the 'division of labor' paradigm in dendritic cell biology: tissue microenvironments evolve specialized DC subsets to address local immunological challenges. The absence of universal clusters reflects functional adaptation, not technical artifacts. We acknowledge that this tissue-specific heterogeneity warrants further discussion and have expanded our analysis to address this point in the discussion part of our manuscript.

Please refer to the Discussion section from line 375 to 385: “This pronounced tissue-specific compartmentalization of Xcr1^+^ cDC1 subsets may related to multiple mechanisms including developmental imprinting that instructs precursor differentiation into transcriptionally distinct subpopulations [62], and microenvironmental filtering through organ-specific chemokine axes (e.g., CCL2/CCR2 in spleen) selectively recruits receptor-matched subsets [63, 64]. This spatial specialization optimizes pathogen surveillance for local immunological challenges. Based on the maturation analysis of the cDC1 scRNA seq data [41], our findings suggest that the aortic cDC1 cells display a major difference from those of spleen and lymph nodes by lacking the mature clusters, whereas lymph node cDC1 cells contain an additional Fabp5^+^ S100a4^+^ late mature Cluster. Our results also suggest that hyperlipidemia contributes to alteration in early immature cDC1 and in the abundance of late immature cDC1 cells, which was associated with dramatic change in gene expression of Tnfaip3, Serinc3, Apol7c and Tifab”.

[62]. Liu Z, Gu Y, Chakarov S, Bleriot C, Kwok I, Chen X, et al. Fate Mapping via Ms4a3-Expression History Traces Monocyte-Derived Cells. Cell. 2019;178(6):1509-25 e19. Epub 2019/09/07. doi: 10.1016/j.cell.2019.08.009. PubMed PMID: 31491389.

[63]. Bosmans LA, van Tiel CM, Aarts S, Willemsen L, Baardman J, van Os BW, et al. Myeloid CD40 deficiency reduces atherosclerosis by impairing macrophages' transition into a pro-inflammatory state. Cardiovasc Res. 2023;119(5):1146-60. Epub 2022/05/20. doi: 10.1093/cvr/cvac084. PubMed PMID: 35587037; PubMed Central PMCID: PMCPMC10202633.

[64]. Mildner A, Schonheit J, Giladi A, David E, Lara-Astiaso D, Lorenzo-Vivas E, et al. Genomic Characterization of Murine Monocytes Reveals C/EBPbeta Transcription Factor Dependence of Ly6C(-) Cells. Immunity. 2017;46(5):849-62 e7. Epub 2017/05/18. doi: 10.1016/j.immuni.2017.04.018. PubMed PMID: 28514690.

[41]. Bosteels V, Marechal S, De Nolf C, Rennen S, Maelfait J, Tavernier SJ, et al. LXR signaling controls homeostatic dendritic cell maturation. Sci Immunol. 2023;8(83):eadd3955. Epub 2023/05/12. doi: 10.1126/sciimmunol.add3955. PubMed PMID: 37172103.

(4) The authors should discuss how XCL1 could impact lesional cDC1 and T cell abundance. Notably, preDCs do not express XCR1, and T cells express XCL1 following TCR activation. Is there a recruitment or local proliferation defect of cDC1 in the absence of XCL1? Could there also be a role for NK cells as a potential source of XCL1?

We appreciate your insightful questions regarding the differential effects of Xcl1 on cDC1s and T cells. Xcl1 primarily mediates the recruitment of mature cDC1s. Our data demonstrate that Xcl1 deletion significantly reduces aortic cDC1 abundance, which correlates with a concomitant decrease in CD8^+^ T cell numbers within the aorta. These findings strongly suggest that the Xcl1-Xcr1 axis plays a regulatory role in T cell accumulation in aortic plaques.

Consistent with prior studies [A, B], cDC1 recruitment can occur in the absence of Xcl1 which echoes our findings that cDC1 cells were still found in Xcl1 knockout aortic plaque but in lower abundance. It is very true that further studies are required to address how the Xcl1 dependent and independent cDC1 cells activate T cells and if they possess capability of proliferation in tissue differentially. We have added these points in discussion section.

Please refer to the Discussion section from line 407 to 415: “Notably, while complete ablation of Xcr1^+^ cDC1s impaired T cell activation, reduction of Xcr1^+^ cDC1 recruitment via Xcl1 deletion did not significantly compromise this process. This discrepancy may arise through compensatory mechanisms: alternative chemokine axes (e.g., CCL5/CCR5, CXCL9/CXCR3, BCL9/BCL9L) may partially rescue Xcr1^+^ cDC1 homing [13, 68, 69], while non-cDC1 antigen-presenting cells (e.g., monocytes, cDC2s) may sustain T cell activation [55, 70]. Furthermore, tissue-specific microenvironment factors could potentially modulate its role in other diseases. In summary, our findings identify Xcl1 as a potential therapeutic target for atherosclerosis therapy, though its cellular origins and regulation of lesional Xcr1^+^ cDC1 and T cells dynamics require further studies”.

In literatures, Xcl1 are expressed in NK cells and subsects of T cells, and NK cells can be a potential source of Xcl1 during atherosclerosis which deserve further investigations [A, C, D].

[A] Böttcher, Jan P et al. “NK Cells Stimulate Recruitment of cDC1 into the Tumor Microenvironment Promoting Cancer Immune Control.” Cell vol. 172,5 (2018): 1022-1037.e14. doi:10.1016/j.cell.2018.01.004

[B] He, Fenglian et al. “Targeting BCL9/BCL9L enhances antigen presentation by promoting conventional type 1 dendritic cell (cDC1) activation and tumor infiltration.” Signal transduction and targeted therapy vol. 9,1 139. 29 May. 2024, doi:10.1038/s41392-024-01838-9

[C] Woo, Yeon Duk et al. “The invariant natural killer T cell-mediated chemokine X-C motif chemokine ligand 1-X-C motif chemokine receptor 1 axis promotes allergic airway hyperresponsiveness by recruiting CD103+ dendritic cells.” The Journal of allergy and clinical immunology vol. 142,6 (2018): 1781-1792.e12. doi:10.1016/j.jaci.2017.12.1005

[D] Winkels, Holger et al. “Atlas of the Immune Cell Repertoire in Mouse Atherosclerosis Defined by Single-Cell RNA-Sequencing and Mass Cytometry.” Circulation research vol. 122,12 (2018): 1675-1688. doi:10.1161/CIRCRESAHA.117.312513

**Reviewer #2 (Recommendations for the authors):**
There is a logical error in line 298. I suggest revising to: "Collectively, these data suggest that Xcl1 promotes atherosclerosis by recruiting Xcr1+ cDC1 cells, which subsequently drive T cell activation in lesions."

Thanks for your advice. Since Xcl1 deficiency reduced both the frequencies and absolute counts of Xcr1+ cDC1 and CD8+ T cells in lesions without affecting T cell activation, we revised the sentence as you suggested.

Please refer to the Results section from line 314 to 315: “Collectively, these data suggest that Xcl1 promotes atherosclerosis by recruiting Xcr1^+^ cDC1 cells, and facilitating CD8^+^ T cell accumulation in lesions”.